# Invertible Monotone Operators for Normalizing Flows

**Byeongkeun Ahn**[1], **Chiyoon Kim**[1], **Youngjoon Hong**[2*], **Hyunwoo J. Kim**[1*]
Korea University[1], Sungkyunkwan University[2]
{byeongkeunahn, kimchiyoon, hyunwoojkim}@korea.ac.kr
hongyj@skku.edu

## Abstract

Normalizing flows model probability distributions by learning invertible transformations that transfer a simple distribution into complex distributions. Since the architecture of ResNet-based normalizing flows is more flexible than that of coupling-based models, ResNet-based normalizing flows have been widely studied in recent years. Despite their architectural flexibility, it is well-known that the current ResNet-based models suffer from constrained Lipschitz constants. In this paper, we propose the *monotone formulation* to overcome the issue of the Lipschitz constants using monotone operators and provide an in-depth theoretical analysis. Furthermore, we construct an activation function called Concatenated Pila (CPila) to improve gradient flow. The resulting model, *Monotone Flows*, exhibits an excellent performance on multiple density estimation benchmarks (MNIST, CIFAR-10, ImageNet32, ImageNet64). Code is available at https://github.com/mlvlab/MonotoneFlows.

## 1 Introduction

Normalizing flows [1, 2] are a method for constructing complex distributions by transforming a probability density through a series of invertible transformations. Normalizing flows are trained using a plain log-likelihood function, and they are capable of exact density evaluation and efficient sampling. Applications include image generation [2, 3, 4, 5, 6, 7, 8], image super-resolution [9, 10, 11], image noise modelling [12, 13], audio synthesis [14, 15], anomaly detection [16, 17], and computational physics [18, 19, 20, 21], highlighting the importance of developing expressive normalizing flows.

However, a major hurdle to designing normalizing flow architectures is that not only should each transformation be invertible but also its Jacobian determinant needs to be tractable [22, 23]. Despite multiple efforts to satisfy the aforementioned constraints [2, 3, 4, 24, 25], the expressive power of the normalizing flow transformations stays limited due to the adoption of specialized architectures [5].

In this regard, ResNet-based normalizing flows [5, 6, 7] are a compelling alternative since they do not impose any structural restrictions. However, they keep the Lipschitz constant of the residual branch less than 1 in order to ensure invertibility. As a result, the Lipschitz constant of each residual block is less than 2. This causes a serious issue on the expressive power since a large Lipschitz constant is often required in converting between distributions. A recent work [26] tackled this problem, but as we show in this paper, their method is equivalent to a simple rescaling of ResNet-based normalizing flows, which has a similar expressive power.

In this work, we propose the approach of *monotone operators* to alleviate the Lipschitz constraint. More precisely, in order to parameterize monotone operators, we use the well-known fact from monotone operator theory: *the Cayley operator of maximally monotone operators (MMOs) is a 1-Lipschitz function*. The residual branch of ResNet-based normalizing flows can be used to parameterize the Cayley operators.

---

*Corresponding authors.

36th Conference on Neural Information Processing Systems (NeurIPS 2022).

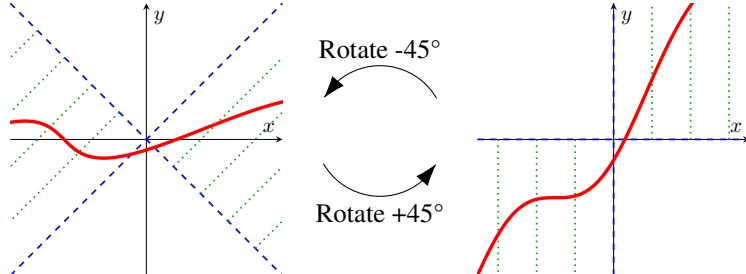

Figure 1: The duality between 1-Lipschitz operators (left) and monotone operators (right) in 1D.

In summary, our contributions are as follows:

- We propose *Monotone Flows*, which greatly loosens the Lipschitz constraint while retaining invertibility and architectural flexibility.
- We derive the *monotone formulation*, a monotone operator-based normalizing flow by parametrizing the Cayley operator, and provide efficient training and inference schemes.
- We propose a new activation function called Pila and its variant Concatenated Pila (CPila) to alleviate the saturated gradient problem.
- We theoretically analyze the expressive power of the monotone formulation and related models [6, 26], and prove our formulation outperforms the other models in practice.
- Monotone Flows consistently outperform comparable baseline normalizing flows on multiple image density estimation benchmarks as well as on 2D toy datasets. In addition, ablation studies demonstrate the effectiveness of the proposed methods.

## 2   Preliminaries

The monotone formulation is a normalizing flow that parametrizes monotone operators in terms of Cayley operators. The parametrization, which must be 1-Lipschitz, and the methods for log determinant computation, are borrowed from ResNet-based normalizing flows. Also, the resulting model is an implicit neural network involving inverse functions. Here, we briefly review these topics.

**Normalizing flows** [1] explicitly model probability distributions by transforming a tractable source distribution, *e.g.*, Gaussian, into a target distribution with a differentiable invertible transform $F : \mathbb{R}^n \to \mathbb{R}^n$. Suppose $F$ maps a sample space $X$ to a latent space $Z$. Given a source distribution $p_Z(z)$, the change-of-variables theorem yields the log-likelihood of any $x \in X$:

$$\log p_X(x) = \log p_Z(z) + \log |\det J_F|, \tag{1}$$

where $z = F(x)$ and $J_F = \partial z/\partial x$ is the Jacobian. Normalizing flows are beneficial since the evaluation of (1) yields the exact density $p_X(x)$. Furthermore, target data $x \sim p_X(x)$ can be efficiently sampled by first sampling $z \sim p_Z(z)$ and then mapping $z$ to $X$ by $x = F^{-1}(z)$. When $F$ consists of multiple transformations applied in succession, each transformation is called a *flow*.

**ResNet-based normalizing flows** have been initially proposed in i-ResNets [5], which are built with a variant of the residual blocks $R(x) = x + G(x)$. By requiring $G(x)$ to be a contraction map, they ensured the invertibility of $R(x)$ with the Banach fixed point theorem. The contractiveness of $G(x)$ is implemented by spectral normalization [27] and 1-Lipschitz activation functions. However, they use a biased truncation of the log Taylor series for computing the log determinant. Residual Flows [6] address this issue with an unbiased estimator based on randomized truncation. They also introduce a new activation function LipSwish to avoid the gradient saturation problem. i-DenseNets [7] improve Residual Flows with a DenseNet variant accompanying learnable concatenation coefficients. They also introduce the Concatenated LipSwish activation function to further improve the gradient flow.

**Implicit neural networks** map an input $x$ to an output $y$ through an implicit equation $F(x, y) = 0$, in contrast with explicit models of the form $y = F(x)$. The main advantage of implicit mappings is their ability to express a richer family of functions under the same parameter budget [28]. Implicit Flows [26] propose to use invertible implicit neural networks in normalizing flows and study an example where residual blocks and the inverse of residual blocks are stacked in an alternating fashion.

**Monotone operator** $F : \mathbb{R}^n \to \mathbb{R}^n$ is an operator that satisfies $\langle u - v, x - y \rangle \geq 0$ for all $x, y \in \mathbb{R}^n$ and all $u \in F(x)$, $v \in F(y)$. [2] This definition generalizes monotonic functions in $\mathbb{R}$, to $\mathbb{R}^n$. The *resolvent* of a monotone operator $F$ is defined as $R_F = (\mathrm{Id} + F)^{-1}$, where $\mathrm{Id}$ is the identity operator. The *Cayley operator* of $F$ is defined as $C_F = 2R_F - \mathrm{Id}$. It is well-known that the Cayley operators of monotone operators are exactly 1-Lipschitz operators. The Cayley operator can be interpreted as a 45-degrees rotation (up to constant scaling) of the corresponding monotone operator in $\mathbb{R}^n \times \mathbb{R}^n$; see Figure 1 for a 1D illustration. Minty surjectivity theorem ([29] and [30, Theorem 21.1]) shows that the Cayley operators of *maximally* monotone operators are exactly 1-Lipschitz *functions*, which are defined at every point. We formally state this correspondence in Theorem 2. See [31] for a survey and [30] for a comprehensive treatment on monotone operators.

## 3 Monotone Flows

In this section, we propose a novel approach, the *monotone formulation*, and describe the training and inference procedure. Moreover, we introduce a new activation function Concatenated Pila (CPila).

### 3.1 Motivation from monotone operator theory

ResNet-based normalizing flows consist of blocks of the form $R(x) = x + G(x)$, where $G(x)$ is a parametrized neural network. If $G(x)$ is a contraction (the Lipschitz constant $\mathrm{Lip}(G) < 1$), the Banach fixed point theorem shows that $R(x)$ is invertible. In this case, $\mathrm{Lip}(R)$ is strictly less than 2, resulting in a severe limitation on the expressive power of the neural network $R$ since a large Lipschitz constant is required in general to transform the source distribution into the target distribution.

However, as shown in the next example, $G(x)$ need not be a contraction for $R(x)$ to be invertible.

**Example 1.** *Consider $G_1(x) = 5x$. Then, $R_1(x) = x + G_1(x) = 6x$ is invertible even though $G_1(x)$ is not a contraction.*

Hence, when considering a relaxed condition and parametrization, it is possible to enhance the expressive power to include a larger class of functions as in Example 1. For this, we introduce monotone operators which have been widely studied in the fields of functional analysis and partial differential equations. We first recall various definitions of the monotonicity of operators in $\mathbb{R}^n$:

**Definition 1.** *Consider an operator $F : \mathbb{R}^n \to \mathbb{R}^n$.*

  (i) *$F$ is* monotone *if for all $x, y \in \mathbb{R}^n$, $\langle F(x) - F(y), x - y \rangle \geq 0$.*
  (ii) *$F$ is* strictly monotone *if for all $x, y \in \mathbb{R}^n$ ($x \neq y$), $\langle F(x) - F(y), x - y \rangle > 0$.*
  (iii) *$F$ is* $\eta$-strongly monotone *($\eta > 0$) if for all $x, y \in \mathbb{R}^n$, $\langle F(x) - F(y), x - y \rangle \geq \eta \|x - y\|_2^2$.*
  (iv) *$F$ is* maximally monotone *if $F$ is monotone and not a proper subset of any monotone operator.*

It is noteworthy that monotonicity does not guarantee injectivity nor surjectivity, and strict monotonicity guarantees injectivity but not surjectivity. However, we have the following theorem:

**Theorem 1.** *An $\eta$-strongly monotone continuous function $F : \mathbb{R}^n \to \mathbb{R}^n$ is invertible for any $\eta > 0$.*

*Proof.* See Corollary 20.28 and Proposition 22.11 in [30].

Theorem 1 easily shows that $R_1(x)$ in Example 1 is invertible. Moreover, if $G(x)$ is $L$-Lipschitz ($L < 1$), $R(x)$ is $(1 - L)$-strongly monotone (see Appendix A.1) and thus invertible by Theorem 1.

As in Example 1, the ResNet-based formulation of normalizing flows does not include all possible invertible and monotone functions. To address this issue, we directly parameterize monotone operators in terms of their Cayley operators. We first formally state the rationale behind our construction.

**Theorem 2.** *Let $F : \mathbb{R}^n \to \mathbb{R}^n$ be an operator and $C_F = 2(\mathrm{Id} + F)^{-1} - \mathrm{Id}$ be its Cayley operator. Then the followings hold.*

  (i) *$F$ is monotone $\Leftrightarrow$ $C_F$ is 1-Lipschitz.*
  (ii) *$F$ is maximally monotone $\Leftrightarrow$ $C_F$ is 1-Lipschitz and $\mathrm{dom}\, C_F = \mathbb{R}^n$.*

*Proof.* See Proposition 23.8 and Proposition 4.4 in [30].

---

[2] We adopt the notation $F(x) = \{y | (x, y) \in F\}$ since $F$ is an operator. However, it suffices to consider single-valued functions defined at every point in our development of theory, in which case the definition reduces to $\langle F(x) - F(y), x - y \rangle \geq 0$.

Theorem 2 shows that each 1-Lipschitz function characterizes a unique MMO since the Lipschitz continuity of $C_F$ implies $C_F$ is single-valued. Hence, the Lipschitz continuous parametrizations of the residual branch in ResNet-based models fit naturally for specifying $C_F$.

We now revisit Example 1 and apply Theorem 2. The Cayley operator of $R_1(x) = 6x$ is given by $C_{R_1}(x) = 2(\mathrm{Id} + R_1)^{-1}(x) - x = -(5/7)x$, which is indeed 1-Lipschitz in accord with Theorem 2.

In general, however, MMOs can be multivalued or undefined at some points. However, when the Cayley operator $C_F$ is an $L$-Lipschitz function with $L < 1$, then $F$ is $(1 - L)/(1 + L)$-strongly monotone and $(1 + L)/(1 - L)$-Lipschitz continuous; see Appendix A.8 for a complete derivation. Theorem 1 then implies $F$ is invertible. Hence, we use $L$-Lipschitz functions for Cayley operators.

### 3.2 Description of the monotone formulation

**Definition 2.** *Let $G : \mathbb{R}^n \to \mathbb{R}^n$ be a Lipschitz-continuous function with Lipschitz constant $L < 1$. The monotone formulation of $G$ is defined as the following function $F : \mathbb{R}^n \to \mathbb{R}^n$:*

$$F(x) = \left( \frac{\mathrm{Id} + G}{2} \right)^{-1}(x) - x, \tag{2}$$

*where* $\mathrm{Id}$ *denotes the identity function.*

Here, $F$ is well-defined since $(\mathrm{Id} + G)/2$ has a unique inverse by the Banach fixed point theorem [5]. The definition of $F$ is a direct application of Theorem 2. On the other hand, from the Cayley operator identity $C_F + C_{F^{-1}} = 0$, we deduce that $F^{-1}$ has the Cayley operator $-G$. Hence, if $y = F(x)$,

$$x = \left( \frac{\mathrm{Id} - G}{2} \right)^{-1}(y) - y, \tag{3}$$

which is well-defined by an analogous argument. Thus we arrive at the following theorem:

**Theorem 3.** *For each $G$ with $\mathrm{Lip}(G) < 1$, there is a unique invertible function $F$ as in* (2).

Note that by rearranging (2) or (3), we obtain the implicit equation connecting $x$ and $y$:

$$x - y = G(x + y). \tag{4}$$

To parametrize $G$, we use the architecture proposed in i-DenseNets [7], but replace the activation function with a new function called Concatenated Pila (CPila), which we introduce later in the paper.

**Remark.** While we parametrize each layer as a monotone operator, the resulting functions need not be monotone. For example, consider the $\pi/3$ counterclockwise rotation operator $R_{\pi/3}$ in $\mathbb{R}^2$. Although $R_{\pi/3}$ is monotone, the composition of $R_{\pi/3}$ with itself is not monotone. This illustrates that monotone functions are not closed under composition, hinting neural networks generated by the composition of invertible monotone operators can represent a much broader class of invertible functions than just monotone operators. Indeed, functions of the monotone formulation are sup-universal diffeomorphism approximators since the monotone formulation includes the "nearly-Id" functions ($\mathcal{R}_L$ in section 4) whose composition suffices for sup-universal diffeomorphism approximation [32].

### 3.3 Training and inference

Training and inference largely follow previous approaches developed in [6, 26] except for minor modifications to adapt them to our monotone formulation. For the function $F$ of Definition 2, implicit differentiation[3] gives

$$\log \det J_F = \mathrm{tr} \left[ \log(I - J_G) - \log(I + J_G) \right], \tag{5}$$

where $J_G = \partial G(w)/\partial w$ at $w = \left( \frac{\mathrm{Id}+G}{2} \right)^{-1}(x)$; see Appendix A.3 for derivation. Now, we construct an unbiased estimator of the log determinant using the Hutchinson trace estimator and the unbiased Russian roulette estimator, following the approach of [6]. Combining (1) and (5),

$$\log p_X(x) = \log p_Z(z) + \mathbb{E}_{n \sim p_N(n), v \sim \mathcal{N}(0,I)} \left[ \sum_{k=1}^{n} \frac{(-1) - (-1)^{k+1}}{k} \frac{v^T J_G^k v}{P(N \geq k)} \right], \tag{6}$$

---

[3]We omit the absolute value sign since the strong monotonicity of $F$ implies $\det J_F > 0$.

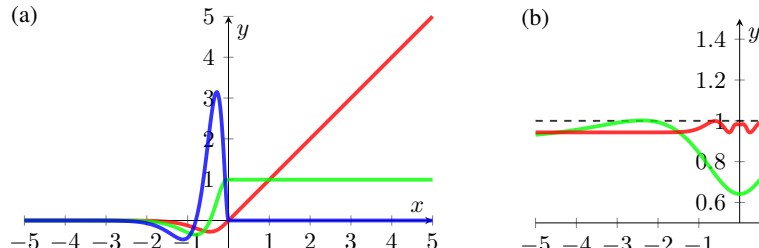

Figure 2: Graphical illustrations of Pila and CPila. (a) The graph of Pila (red) and its first (green) and second derivatives (blue) with $k = 5$. (b) The speed of the curve of CPila (red) with $k = 5$ and CLipSwish (green) with $\beta = 1$.

where $p_N(n)$ is a distribution with support over the positive integers. Note that the derivative of the log-likelihood (6) can be evaluated by the memory-efficient Neumann gradient estimator as in [6].

Training consists of forward and backward propagations using (2) and (6). For forward propagation, $F$ can be evaluated with fixed-point iterations since $G$ is a contraction mapping while it suffices to backpropagate through $(\mathrm{Id} + G)^{-1}$ since $F(x) = \left(\frac{\mathrm{Id}+G}{2}\right)^{-1}(x) - x = (\mathrm{Id} + G)^{-1}(2x) - x$. For this, we follow the approach of [26]. Let $G$ be parametrized by $\theta$. Denote the training objective (6) as $\ell$ and write $w = (\mathrm{Id} + G)^{-1}(u)$ (where $u = 2x$). Implicit differentiation yields

$$\frac{\partial \ell}{\partial u} = \frac{\partial \ell}{\partial w}(I + J_G)^{-1}, \quad \frac{\partial \ell}{\partial \theta} = \left(\frac{\partial \ell}{\partial w}(I + J_G)^{-1}\right)\frac{\partial G}{\partial \theta}, \tag{7}$$

where $G$ and $J_G$ are evaluated at $w = (\mathrm{Id} + G)^{-1}(u)$; see Appendix A.5 for derivation. Since the linear map $J_G : \mathbb{R}^n \to \mathbb{R}^n$ is a contraction mapping, the vector-matrix inverse product in (7) can be evaluated by a fixed-point iteration. For this, we adopt the secant method, but when convergence is not improved for 10 consecutive iterations, the Krasnoselskii-Mann iteration is used instead.

Inference consists of forward propagation through (3), which is almost the same as the forward propagation in training except that the sign of $G$ is the opposite.

### 3.4 Concatenated positive identity 1-Lipschitz activation function

The LipSwish activation function proposed in [6] is based on the premise that having a non-vanishing second-order gradient would improve training. However, LipSwish still suffers from the vanishing gradient problem since its first-order derivative stays around 0.5 in the neighborhood of $x = 0$ where most of the preactivations would reside. Concatenated LipSwish (CLipSwish) [7] improves upon LipSwish by concatenating $\mathrm{LipSwish}(x)$ and $\mathrm{LipSwish}(-x)$, but a similar issue still occurs as shown in Figure 2 (b).

In this regard, we propose Pila (positive identity 1-Lipschitz activation) function, which makes the gradient flow equal to the identity function on $x \geq 0$ during backpropagation. When $x < 0$, Pila converges to zero as $x \to -\infty$ while matching the identity function at $x = 0$ up to the third derivative. The Pila function is described as follows:

$$\mathrm{Pila}(x) = \begin{cases} x & \text{if } x \geq 0, \\ \left(\frac{k^2}{2}x^3 - kx^2 + x\right)e^{kx} & \text{if } x < 0. \end{cases} \tag{8}$$

Here, $k > 0$ is treated either as a fixed hyperparameter or a learnable parameter. In all of our experiments, we set $k = 5$ and do not learn $k$. Note that $\mathrm{Lip}(\mathrm{Pila}) = 1$ for any $k > 0$.

As in [7], we introduce a concatenated version of Pila, which we call Concatenated Pila (CPila):

$$\mathrm{CPila}(x) = \alpha_1[\mathrm{Pila}(x - \alpha_2), \mathrm{Pila}(-x - \alpha_2)]^T, \tag{9}$$

where $\alpha_1 = 1/1.06$ and $\alpha_2 = 0.2$. By rescaling and translating with $\alpha_1$ and $\alpha_2$, we fabricate the CPila function to be 1-Lipschitz. Figure 2 (b) describes the *speed* of the curve[4], which shows CPila has a speed closer to 1 around $x = 0$ compared to the speed of CLipSwish.

---

[4]speed of the curve at time $t$: $|v(t)| = \sqrt{(x'(t))^2 + (y'(t))^2}$

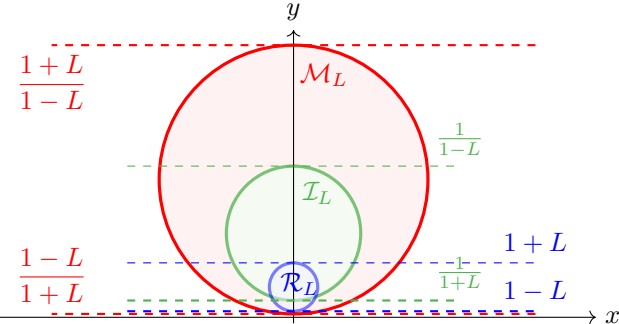

Figure 3: Comparison of $\mathcal{R}_L$ (blue), $\mathcal{I}_L$ (green), and $\mathcal{M}_L$ (red). The figure depicts the possible ranges of $q = F(x_1) - F(x_2)$ with respect to $p = x_1 - x_2$ for each function class, assuming the domain and the codomain of $F$ are both $\mathbb{R}^2$. For visualization, we choose $L = 0.8$ and fix $p$ as a unit vector pointing in the $+y$ direction.

## 4 Expressive power

In this section, we analyze the expressive power of the monotone formulation and related models. We prove that the monotone formulation is superior to the residual formulation [6] and the implicit formulation of Lu et al. [26] when the Lipschitz constant $L$ of the $G$-network is less than unity. In practice, since $L$ should not be close to unity, the monotone formulation is a better choice than the residual formulation or the implicit formulation of Lu et al.

We start with a formal definition of the function classes of each model.

**Definition 3.** *For $0 \leq L < 1$, we define the following function classes:*

| | |
|---|---|
| $L$-Lipschitz functions | $\mathcal{G}_L = \left\{ G \in C^2(\mathbb{R}^n, \mathbb{R}^n) \| \mathrm{Lip}(G) = L \right\}$ |
| Residual formulation | $\mathcal{R}_L = \left\{ \mathrm{Id} + G \| G \in \mathcal{G}_L \right\}$ |
| Inverse residual formulation | $\mathcal{I}_L = \left\{ (\mathrm{Id} + G)^{-1} \| G \in \mathcal{G}_L \right\}$ |
| Monotone formulation | $\mathcal{M}_L = \left\{ \left( \frac{\mathrm{Id} + G}{2} \right)^{-1} - \mathrm{Id} \| G \in \mathcal{G}_L \right\}$ |

*Here, $C^2(\mathbb{R}^n, \mathbb{R}^n)$ denotes the set of twice continuously differentiable functions.[5]*

Note that the implicit formulation of Lu et al. [26] defines one block as the composition of two functions $F_2^{-1} \circ F_1$ for any $F_1, F_2 \in \mathcal{R}_L$. We perform our analysis only on the second layers, which corresponds to the comparison between $\mathcal{R}_L$ and $\mathcal{I}_L$. For convenience, we use the following notation.

**Definition 4.** *Let $A$ be one of $\mathcal{G}_L, \mathcal{R}_L, \mathcal{I}_L,$ or $\mathcal{M}_L$, and $\alpha > 0$. Then,*

$$\alpha A = \left\{ \alpha F | F \in A \right\}, \quad \mathbb{R}^+ A = \left\{ \beta F | \forall \beta \in \mathbb{R}^+, F \in A \right\}.$$

We now examine the relations between the function spaces.

**Theorem 4.** *For $0 \leq L < 1$, the following holds.*

$$(i)\, \mathcal{I}_L = \frac{1}{1 - L^2} \mathcal{R}_L, \quad (ii)\, \mathcal{M}_L = \frac{1 + L^2}{1 - L^2} \mathcal{R}_{\frac{2L}{1+L^2}}, \quad (iii)\, \mathcal{R}_L \subsetneq \mathcal{M}_L, \quad (iv)\, \mathcal{I}_L \subsetneq \mathcal{M}_L.$$

*Proof.* See Appendix A.6 for a complete proof.

Figure 3 shows the ranges of $q = F(x_1) - F(x_2)$ with respect to $p = x_1 - x_2$ for $\mathcal{R}_L, \mathcal{I}_L,$ and $\mathcal{M}_L$, clearly demonstrating the relations (i)-(iv) of Theorem 4.

Theorem 4 shows the inverse residual formulation $\mathcal{I}_L$ can only express functions that are a constant multiple of the functions expressed by the residual formulation $\mathcal{R}_L$. Hence, the inverse residual formulation has no advantage over the residual formulation in this respect. On the other hand, the monotone formulation $\mathcal{M}_L$ can express functions that are not expressible by any constant multiple of the functions in $\mathcal{R}_L$, because $2L/(1 + L^2) > L$ whenever $L < 1$.

When $L$ can approach unity arbitrarily close, the monotone formulation has the same expressive power as the residual formulation or the inverse residual formulation. However, this is not possible

---

[5]We require continuous second-order derivatives as the gradient of the log Jacobian determinant is of second-order. However, the results still hold when a weaker function class is considered (e.g., $C^1(\mathbb{R}^n, \mathbb{R}^n)$).

in practice because the variance of the log determinant estimator (6) diverges to infinity when $L$ is greater than a specific threshold less than unity that depends on the distribution $p_N(n)$ [33]. Since the log determinant estimator (6) is a linear combination of the log determinant estimator of the same $G$-network's residual formulation, (6) has a finite variance whenever the corresponding residual formulation's estimator possesses a finite variance. Suppose the constraints, in practice, bound the Lipschitz constant of the $G$-networks to be at most $L_{\max} < 1$. Then, the residual formulation gives rise to $\mathbb{R}^+ \mathcal{R}_{L_{\max}}$ whereas the monotone formulation provides $\mathbb{R}^+ \mathcal{M}_{L_{\max}} = \mathbb{R}^+ \mathcal{R}_{\frac{2L_{\max}}{1+L_{\max}^2}} \supsetneq \mathbb{R}^+ \mathcal{R}_{L_{\max}}$.

An analogous argument holds for $\mathbb{R}^+ \mathcal{I}_{L_{\max}}$. Hence, the analysis shows that the monotone formulation outperforms the other formulations in terms of the expressive power.

## 5  Related work

**Normalizing flows.** Normalizing flows require each layer to be invertible and the Jacobian determinant of each layer to be tractable, which have spurred multiple approaches for architecture design. Restricted Jacobian models introduce specific structures to the Jacobian to make the Jacobian determinants tractable. They include determinant identity-based models [1, 25] which use the Weinstein–Aronszajn identity with reduced intermediate layer dimensions, and coupling-based models [2, 3, 4, 34, 35] which split the input into two parts and transform one part conditioned on another. However, the restrictions on the Jacobian limit their expressivity. In contrast, free-form Jacobian models are not subject to the same constraints and include ResNet-based models (discussed in section 2) and Neural ODEs. Neural ODEs [36] parametrize the rate of change in ODEs with neural networks. They ensure invertibility through the existence and uniqueness theorem of ODEs. However, they have relatively high time complexity since they require solving ODEs numerically on every iteration.

So far, the normalizing flows we have discussed are non-autoregressive models. In contrast, autoregressive models [37, 38, 39] decompose probability distributions as conditional distributions depending only on the previous values. While autoregressive models typically achieve higher log-likelihoods, they are orders of magnitudes slower to train or sample [4]. Therefore, we focus on non-autoregressive models in our paper.

**Monotone operators in neural networks.** [40] pioneers monotone operator neural networks to assure the convergence of fixed-point iterations in *implicit depth* networks. They build linear monotone operators by parametrizing the symmetric and antisymmetric parts separately. Positive defininteness is enforced only on the symmetric part. One shortcoming is that the construction is limited to *linear* monotone operators, which our work addresses using nonlinear Cayley operators. [41] proposes the first neural network model of nonlinear MMOs by parametrizing the Cayley operators. One key difference is that the 1-Lipschitz constraint on Cayley operators is enforced through a penalty loss, which may fail to control the Lipschitz constant globally [5]. Our method systematically ensures 1-Lipschitzness by the use of spectral normalization [27] and 1-Lipschitz activations, following [5, 6, 7]. Also, their work does not consider normalizing flows. To the best of our knowledge, our work proposes the first normalizing flow based on monotone operators.

## 6  Experiments

We start by validating the modeling capacity of our model with a 1D toy function. We then evaluate Monotone Flows on density estimation of 2D toy and image datasets and conduct ablation studies. In all experiments, we parameterize the $G$-network in Definition 2 using i-DenseNets [7] with our activation function CPila (except for the 1D toy experiments that use CReLU and for part of the ablation study where we ablate CPila). The $G$-network consists of fully-connected layers for 1D and 2D toy tasks and convolutional layers for image tasks. We use the Poisson distribution for $p_N(n)$ whenever the estimator (6) is used. We use Adam [42] for all experiments.

### 6.1  Toy 1D experiments

Figure 4 illustrates the expressive power of the monotone formulation with a 1D toy example. We fit a 1D function that consists of four consecutive step-like shapes designed to exhibit a high Lipschitz constant. We consider four different 2-layer networks, as shown in Figure 4. Experimental details

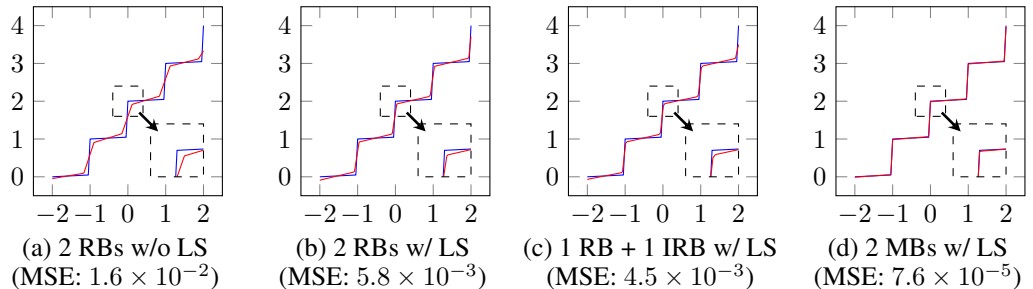

| (a) 2 RBs w/o LS | (b) 2 RBs w/ LS | (c) 1 RB + 1 IRB w/ LS | (d) 2 MBs w/ LS |
|:---:|:---:|:---:|:---:|
| (MSE: $1.6 \times 10^{-2}$) | (MSE: $5.8 \times 10^{-3}$) | (MSE: $4.5 \times 10^{-3}$) | (MSE: $7.6 \times 10^{-5}$) |

Figure 4: Comparison of 2 RBs without learnable scaling, 2 RBs, 1 RB followed by 1 IRB, and 2 MBs. All experiments except (a) are performed with learnable scaling (LS). Blue and red lines represent the target function and the approximation by neural networks, respectively.

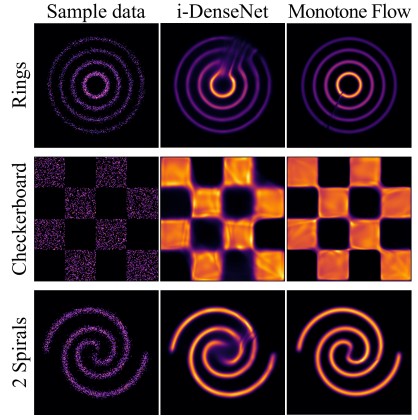

Figure 5: 2D toy density modeling results (full results in Appendix D).

Table 1: Toy density modelling results in nats. We display the average of the test loss for the last 20 tests at checkpoints (iterations 48100, 48200, ..., 50000) for a single run.

| Data | i-DenseNet | Monotone Flow |
|---|---|---|
| 2 Spirals | 2.729 | **2.658** |
| 8 Gaussians | **2.840** | **2.840** |
| Checkerboard | 3.609 | **3.540** |
| Circles | 3.280 | **3.276** |
| Moons | 2.401 | **2.400** |
| Pinwheel | 2.343 | **2.333** |
| Rings | 2.884 | **2.665** |
| Swissroll | 2.680 | **2.676** |

are discussed in Appendix C.1. We denote residual block, inverse residual block, and monotone block by RB, IRB, and MB. Note that the learnable scaling (LS) is used only for Figure 4 (b)-(d). The $G$-networks in the experiments (a)-(d) have the same structure, and hence the same number of parameters are used. Consequently, differences in the result solely come from the differences in the formulations. Figure 4 (a) and (b) demonstrate that the expressive power is improved when using $\mathbb{R}^{+}\mathcal{R}_{L}$ instead of $\mathcal{R}_{L}$. Figure 4 (b) and (c) reveal that the expressive power of $\mathbb{R}^{+}\mathcal{R}_{L}$ and $\mathbb{R}^{+}\mathcal{I}_{L}$ are similar, in agreement with Theorem 4. Figure 4 (d) shows that our monotone formulation $\mathbb{R}^{+}\mathcal{M}_{L}$ exceedingly outperforms the other networks.

## 6.2 Density estimation on 2D toy data

We use the 2D toy datasets provided with the official source code of i-DenseNets [7]. Following [7], we use ten flow blocks (see Appendix C.2 for details). Exact Jacobian evaluation is used instead of the stochastic estimation (6) as it is inexpensive for 2D data. We train the models for 50K epochs with the learnable concatenation of i-DenseNets enabled after 25K epochs. We use a batch size of 500.

Qualitative and quantitative results in Figure 5 and Table 1 show that Monotone Flows outperform i-DenseNets in almost all settings, especially on challenging datasets. One limitation of normalizing flows manifests in the qualitative results. For instance, in the dataset 'rings', the circles cannot be closed because normalizing flows are required to preserve topology. Despite the inherent restriction, our model finds an excellent approximation.

## 6.3 Density estimation on images

We evaluate our method on MNIST [43], CIFAR-10 [44], and the downscaled version of ImageNet in 32×32 and 64×64 [45].[6] We use (6) for estimating the Jacobian determinant. We use uniform

---

[6]There are two versions of the downscaled ImageNet. The one used for evaluating normalizing flow models is the version packed in *tar*. It has been removed from the official website but is still available through other routes (e.g., Academic Torrents).

Table 2: Density estimation results on images in bits-per-dimension (bpd) with the number of parameters of each model. All numbers except for the last row are with uniform dequantization. VDQ: variational dequantization.

| | MNIST | | CIFAR-10 | | ImageNet32 | | ImageNet64 | |
|---|---|---|---|---|---|---|---|---|
| Model | bpd ↓ | params | bpd ↓ | params | bpd ↓ | params | bpd ↓ | params |
| Real NVP [3] | 1.06 | - | 3.49 | 6.4M | 4.28 | 46.0M | 3.98 | 96.0M |
| Glow [4] | 1.05 | - | 3.35 | 44.2M | 4.09 | 66.1M | 3.81 | 111.1M |
| FFJORD [36] | 0.99 | - | 3.40 | - | - | - | - | - |
| i-ResNet [5] | 1.06 | - | 3.45 | 44.2M | - | - | - | - |
| Residual Flow [6] | 0.97 | 16.6M | 3.28 | 25.2M | 4.01 | 47.1M | 3.76 | 53.3M |
| i-DenseNet [7] | - | - | 3.25 | 24.9M | 3.98 | 47.0M | - | - |
| Monotone Flow | **0.928** | 20.9M | **3.215** | 24.9M | **3.961** | 47.0M | **3.734** | 48.9M |
| Monotone Flow + VDQ | - | - | **3.062** | 46.9M | **3.901** | 69.0M | - | - |

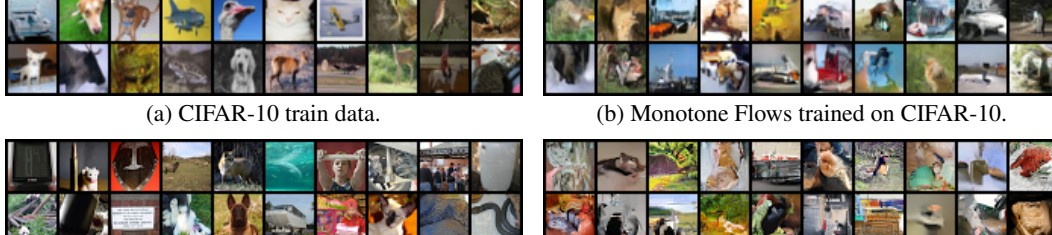

(a) CIFAR-10 train data.   (b) Monotone Flows trained on CIFAR-10.

(c) ImageNet32 train data.   (d) Monotone Flows trained on ImageNet32.

Figure 6: Train data and generated samples of CIFAR-10 and ImageNet32.

dequantization for both training and testing. For MNIST and CIFAR-10, the learnable concatenation is enabled after 25 epochs; for ImageNet32 and ImageNet64, it is enabled from the start. We train for 100, 1,000, 20, 20 epochs with batch sizes 64, 64, 256, 256 with learning rates 0.001, 0.001, 0.004, 0.004 for MNIST, CIFAR-10, ImageNet32, ImageNet64, respectively. We report single-seed results following [4, 5, 6, 7]. Details are in Appendix C.3. Samples from the trained models are displayed in Figure 6. More samples can be found in Appendix E.

Table 2 shows our model outperforms baselines on all datasets considered, demonstrating the effectiveness of our proposal. In concrete numbers, our model reduces 0.042 bpd in MNIST, 0.035 bpd in CIFAR-10, 0.019 bpd in ImageNet32, and 0.023 bpd in ImageNet64 compared to baseline normalizing flow models.

**Remark on comparison.** Flow++ [34] introduced variational dequantization which improves the log-likelihood, reporting 3.08 and 3.86 bits/dim on CIFAR-10 and ImageNet32, respectively. With variational dequantization, Monotone Flows achieve 3.062 and 3.901 bits/dim on the same benchmarks. Also, there is a line of research that achieves better likelihood values by introducing latent variables in extra dimensions, at the cost of losing exact likelihood computation [46, 47, 48]. On the other hand, ScoreFlow [49] achieves better likelihood values by training with a weighted score-matching objective that lower bounds the log likelihood. We leave it for future research to adapt these methods for Monotone Flows.

### 6.4 Ablation study

Our model has two new components: the monotone formulation and CPila activation function. We quantify their contributions by ablating either of the two components while leaving the other intact. When ablating the monotone formulation (row #2), we used the residual connections of [7]; when ablating CPila (row #3), we used CLipSwish instead. Table 3 compares the performance of i-DenseNets (row #1) and the ablated models on the CIFAR-10 dataset under the same experimental setup as in section 6.3. Details are in Appendix C.5. Results confirm that both components are

Table 3: Ablation study for density estimation on CIFAR-10. MFL: monotone formulation.

| # | MFL | CPila | bpd ↓ |
|---|---|---|---|
| 1 | ✘ | ✘ | 3.252 |
| 2 | ✘ | ✔ | 3.243 |
| 3 | ✔ | ✘ | 3.229 |
| 4 | ✔ | ✔ | 3.215 |

beneficial for achieving a low bits-per-dimension (bpd). The monotone formulation provides a higher performance gain (0.028 bpd) than CPila (0.014 bpd).

## 7 Conclusion

We presented Monotone Flows, a novel parametrization of normalizing flows based on monotone operators (the monotone formulation) combined with a new activation function called Concatenated Pila (CPila). Our theoretical analysis elucidated why the monotone formulation is more expressive than the residual or inverse residual formulations. Experimental results demonstrated the effectiveness of the proposed method on various density estimation benchmarks. We believe our contribution is a solid step towards improving the expressivity of normalizing flows.

## Acknowledgments and Disclosure of Funding

Our work is supported by ICT Creative Consilience program (IITP-2022-2020-0-01819) supervised by the IITP, and Kakao Brain Corporation. The work of Y. Hong was supported by Basic Science Research Program through the National Research Foundation of Korea (NRF) funded by the Ministry of Education (NRF-2021R1A2C1093579) and the Korea government (MSIT) (No. 2022R1A4A3033571).

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
