# Invertible Monotone Operators for Normalizing Flows (Supplementary Material)

**Overview.** The Appendix is organized as follows. In Appendix A, we present the proofs of the theorems and lemmas stated in the main text. In Appendix B, we present the forward and backward algorithms for a single monotone layer. In Appendix C, we describe the experimental details for each experiment discussed in the main text and present the classification experiment on CIFAR-10. In Appendix D and E, we include the visualizations of the 2D toy experiments and the generated images from our trained model. In Appendix F, we discuss the limitations and potential negative societal impacts of our work.

## A  Proofs and derivations

### A.1  Proof of the strong monotonicity of a single residual block

We prove that $R(x) = x + G(x)$ with $\mathrm{Lip}(G) = L$ is $(1-L)$-strongly monotone. A direct calculation shows

$$\begin{aligned}
\langle R(x) - R(y), x - y \rangle &= \|x - y\|_2^2 - \langle G(x) - G(y), x - y \rangle \\
&\geq \|x - y\|_2^2 - \|G(x) - G(y)\|_2 \|x - y\|_2 \\
&\geq \|x - y\|_2^2 - L\|x - y\|_2\|x - y\|_2 = (1 - L)\|x - y\|_2^2.
\end{aligned}$$

### A.2  Proof of Theorem 2

As stated in the main text, a complete proof can be found in [30, Proposition 23.8 and Proposition 4.4]. Here, we provide an alternative proof, largely based on the proof of Alberti and Ambrosio [50], to keep our paper self-contained. Before proving Theorem 2, we state Kirszbraun's theorem [51].

**Theorem 5.** *(Kirszbraun, 1934) If $U \subseteq \mathbb{R}^n$ and $f : U \to \mathbb{R}^m$ is $K$-Lipschitz, there is a $K$-Lipschitz function $g : \mathbb{R}^n \to \mathbb{R}^m$ which is an extension of of $f$ to $\mathbb{R}^n$.*

Now we prove Theorem 2.

*Proof.* (i) ($\Rightarrow$) Assume $F$ is monotone, and let $(x_1, y_1), (x_2, y_2) \in C_F$. By rearranging the equation of $C_F$, we obtain

$$b_i = F(a_i), \quad a_i = \frac{x_i + y_i}{2}, \quad b_i = \frac{x_i - y_i}{2}, \quad \text{for } i = 1, 2.$$

Hence, we find that

$$\begin{aligned}
\|y_1 - y_2\|_2^2 &= \|(a_1 - b_1) - (a_2 - b_2)\|_2^2 \\
&= \|(a_1 - a_2) - (b_1 - b_2)\|_2^2 \\
&= \|a_1 - a_2\|_2^2 + \|b_1 - b_2\|_2^2 - 2\langle a_1 - a_2, b_1 - b_2 \rangle \\
&\leq \|a_1 - a_2\|_2^2 + \|b_1 - b_2\|_2^2 + 2\langle a_1 - a_2, b_1 - b_2 \rangle \\
&= \|(a_1 - a_2) + (b_1 - b_2)\|_2^2 \\
&= \|(a_1 + b_1) - (a_2 + b_2)\|_2^2 \\
&= \|x_1 - x_2\|_2^2,
\end{aligned}$$

where the inequality follows from the monotonicity of $F$. Hence, $C_F$ is 1-Lipschitz.

($\Leftarrow$) Assume $C_F$ is 1-Lipschitz, and let $(x_1, y_1), (x_2, y_2) \in F$. Rearranging the equation of $C_F$, we deduce that

$$d_i = C_F(c_i), \quad c_i = x_i + y_i, \quad d_i = x_i - y_i, \quad \text{for } i = 1, 2.$$

Hence, we derive that

$$\langle y_1 - y_2, x_1 - x_2 \rangle = \left\langle \frac{c_1 - d_1}{2} - \frac{c_2 - d_2}{2}, \frac{c_1 + d_1}{2} - \frac{c_2 + d_2}{2} \right\rangle$$

$$= \left\langle \frac{c_1 - c_2}{2} - \frac{d_1 - d_2}{2}, \frac{c_1 - c_2}{2} + \frac{d_1 - d_2}{2} \right\rangle$$

$$= \frac{1}{4}\left( \|c_1 - c_2\|_2^2 - \|d_1 - d_2\|_2^2 \right) \geq 0,$$

where the last inequality follows from the 1-Lipschitz condition of $C_F$. Hence, $F$ is monotone.

(ii) ($\Rightarrow$) Assume $F$ is maximally monotone. Then, $C_F$ is 1-Lipschitz by (i), whence $C_F$ is at most single-valued at each point. Suppose $\operatorname{dom} C_F \subsetneq \mathbb{R}^n$. By Kirszbraun's theorem (Theorem 5), there is an extension $F_e$ of $C_F$ such that $F_e$ is 1-Lipschitz and $\operatorname{dom} F_e = \mathbb{R}^n$. Since $F_e$ is 1-Lipschitz, the operator $\tilde{F}$ that has $F_e$ as its Cayley operator is monotone. We have $F \subsetneq \tilde{F}$, which contradicts the maximality of $F$.

($\Leftarrow$) Assume $C_F$ is 1-Lipschitz and $\operatorname{dom} C_F = \mathbb{R}^n$. Then, $F$ is monotone by (i). Suppose $F$ is not maximally monotone. This implies there exists an $(x', y') \in \mathbb{R}^n \times \mathbb{R}^n$ such that $(x', y') \notin F$ but $\widehat{F} := F \cup \{(x', y')\}$ is monotone. Thus, the Cayley operator of $\widehat{F}$,

$$C_{\widehat{F}} = C_F \cup \{(x' + y', x' - y')\},$$

is 1-Lipschitz by (i). On the other hand, $C_{\widehat{F}}$ has a function value $C_F(x' + y')$ at $x' + y'$. This does not equal $x' - y'$ since $(x', y') \notin F$. This means $C_{\widehat{F}}$ is multi-valued at $x' + y'$, which contradicts $C_{\widehat{F}}$ is 1-Lipschitz. ∎

### A.3 Derivation of equation (5)

As mentioned in the main text, we assume $G$ is continuously twice differentiable and is a contraction mapping with $\operatorname{Lip}(G) = L < 1$. The Jacobian of $F$ can be calculated in a straightforward manner from the explicit form (2):

$$J_F = \frac{\partial y}{\partial x} = J_{\left(\frac{\mathrm{Id}+G}{2}\right)^{-1}} - I = 2J_{(\mathrm{Id}+G)^{-1}} - I.$$

Here, we have $\|J_G\|_2 \leq \operatorname{Lip}(G) = L$, where $\|\cdot\|$ denotes the matrix spectral norm. Thus, for any $v \in \mathbb{R}^n$ with $\|v\|_2 = 1$, we have $v^T J_{\mathrm{Id}+G} v = v^T v + v^T J_G v \geq 1 - L > 0$. This implies $J_{\mathrm{Id}+G}$ is not singular and thus invertible. By inverse function theorem, we obtain that $J_{(\mathrm{Id}+G)^{-1}} = J_{\mathrm{Id}+G}^{-1}$. Therefore, we deduce that

$$J_F = 2J_{(\mathrm{Id}+G)^{-1}} - I$$
$$= 2J_{\mathrm{Id}+G}^{-1} - I$$
$$= 2(I + J_G)^{-1} - I$$
$$= (I + J_G)^{-1}(2I - (I + J_G))$$
$$= (I + J_G)^{-1}(I - J_G),$$

where $J_G$ is evaluated at $w = (\mathrm{Id} + G)^{-1}(u)$ and $u = 2x$. Hence, we find that

$$\log \det J_F = \log \det \left[ (I + J_G)^{-1}(I - J_G) \right]$$
$$= \log \det(I - J_G) - \log \det(I + J_G)$$
$$= \operatorname{tr}\left[ \log(I - J_G) - \log(I + J_G) \right]$$

where the function $\log$ in the last line denotes the matrix logarithm (not an elementwise logarithm). In our implementation, we use $G(x) = -2H(x/\sqrt{2})$ where $H$ is an $L$-Lipschitz function. This change does not affect the Lipschitz constant in the calculation, and transforms the formulation of $F$ to $F(x) = \sqrt{2}(\mathrm{Id} - H)^{-1}(\sqrt{2}x) - x$. The inverse of $F$ can be derived by $F^{-1}(x) = \sqrt{2}(\mathrm{Id} + H)^{-1}(\sqrt{2}x) - x$ as in (3).

## A.4 Derivation of equation (6)

By Taylor expansion, we derive that

$$\log \det J_F = \operatorname{tr}\left[\log(I - J_G) - \log(I + J_G)\right] = \operatorname{tr}\left[\sum_{k=1}^{\infty} \frac{(-1) - (-1)^{k+1}}{k} J_G^k\right].$$

Here, Taylor expansion is justified because $||J_G||_2 \leq \operatorname{Lip}(G) = L < 1$. Now, with neural networks, the exact evaluation of $J_G^k$'s is not feasible due to computational complexity. Instead, we resort to an estimator called the Hutchinson trace estimator, which only requires the vector-matrix or matrix-vector product. Given a general matrix $A \in \mathbb{R}^{n \times n}$, the Hutchinson trace estimator is defined as

$$\operatorname{tr}(A) = \mathbb{E}_{v \sim \mathcal{N}(0, I)}\left[v^T A v\right],$$

where $v$ is sampled from a multivariate standard normal distribution. When applied to our case, this yields

$$\log \det J_F = \mathbb{E}_{v \sim \mathcal{N}(0, I)}\left[\sum_{k=1}^{\infty} \frac{(-1) - (-1)^{k+1}}{k} v^T J_G^k v\right].$$

The calculation contains an infinite series whose evaluation is difficult. However, the infinite sum can be approximated by a finite number of terms using the unbiased Russian roulette estimator:

$$\log \det J_F = \mathbb{E}_{n \sim p_N(n), v \sim \mathcal{N}(0, I)}\left[\sum_{k=1}^{n} \frac{(-1) - (-1)^{k+1}}{k} \frac{v^T J_G^k v}{P(N \geq k)}\right].$$

Here, the distribution $p_N(n)$ can be any distribution with $P(N \geq k) > 0$ for all natural numbers $k \in \mathbb{N}$; for instance, a geometric distribution is chosen in [6], and a Poisson distribution is chosen in [7]. In our case, we use the Poisson distribution following [7].

The Russian roulette estimator is justified by Lemma 3 in Appendix B of [6]. More precisely, we deduce that

$$\sum_{k=1}^{\infty}\left|\frac{(-1) - (-1)^{k+1}}{k} v^T J_G^k v\right| \leq 2 \sum_{k=1}^{\infty}\left|\frac{1}{k} v^T J_G^k v\right|$$

$$\leq 2 \sum_{k=1}^{\infty} \frac{1}{k}\left\|v\right\|_2 \left\|J_G\right\|_2^k \left\|v\right\|_2$$

$$= 2 \left\|v\right\|_2^2 \sum_{k=1}^{\infty} \frac{1}{k}\left\|J_G\right\|_2^k$$

$$\leq 2 \left\|v\right\|_2^2 \sum_{k=1}^{\infty} \frac{1}{k}\operatorname{Lip}(G)^k$$

$$= 2 \left\|v\right\|_2^2 \log(1 - \operatorname{Lip}(G)) < \infty.$$

We include the following lemma here to keep our paper self-contained.

**Lemma 2.** *(Lemma 3 in [6]) (Unbiased randomized truncated series) Let $Y_k$ be a real random variable with $\lim_{k \to \infty} \mathbb{E}[Y_k] = a$ for some $a \in \mathbb{R}$. Further, let $\Delta_0 = Y_0$ and $\Delta_k = Y_k - Y_{k-1}$ for $k \geq 1$. Assume $\mathbb{E}\left[\sum_{k=0}^{\infty}|\Delta_k|\right] < \infty$ and let $N$ be a random variable with support over the positive integers and $n \sim p_N(n)$. Then for*

$$Z = \sum_{k=0}^{n} \frac{\Delta_k}{P(N \geq k)},$$

*we find*

$$\lim_{k \to \infty} \mathbb{E}[Y_k] = \mathbb{E}_{n \sim p_N(n)}[Z] = a.$$

In our case, $\Delta_0 = 0$ and for $k \geq 1$

$$\Delta_k = \frac{((-1) - (-1)^{k+1})}{k} v^T J_G^k v.$$

## A.5 Derivation of equation (7)

Backward propagation is done through implicit differentiation by adapting the formulation in [26]. For completeness, we include the derivation here.

Since $F(x) = \left(\frac{\mathrm{Id}+G}{2}\right)^{-1}(x) - x = (\mathrm{Id} + G)^{-1}(2x) - x$, it suffices to backpropagate through the function $(\mathrm{Id} + G)^{-1}$. Let $w = (\mathrm{Id} + G)^{-1}(u)$ (with $u = 2x$), where $G$ is parameterized with parameters $\theta$. Here, $u$ and $\theta$ are independent variables, and $w$ and the loss $\ell$ are dependent variables. We first note that by chain rule

$$\frac{\partial \ell}{\partial u} = \frac{\partial \ell}{\partial w} \frac{\partial w}{\partial u}, \quad \frac{\partial \ell}{\partial \theta} = \frac{\partial \ell}{\partial w} \frac{\partial w}{\partial \theta}.$$

Since $\partial\ell/\partial y$ is given by backpropagation, we only need to estimate the vector-Jacobian products $(\partial\ell/\partial w)(\partial w/\partial u)$ and $(\partial\ell/\partial w)(\partial w/\partial\theta)$ from $\partial\ell/\partial y$, not the full Jacobians $\partial w/\partial u$ and $\partial w/\partial\theta$.

To find $\partial\ell/\partial u$, we consider the implicit equation between $u$ and $w$ given by

$$w + G(w, \theta) - u = 0.$$

Taking a derivative with respect to $w$ while holding $\theta$ constant yields

$$\frac{\partial w}{\partial u} + J_G \frac{\partial w}{\partial u} - I = 0,$$

where $J_G \equiv (\partial G(x, \theta)/\partial x)|_{x=w,\,\theta=\theta}$. Hence, we obtain

$$\frac{\partial \ell}{\partial u} = \frac{\partial \ell}{\partial w}(I + J_G)^{-1} \quad \Rightarrow \quad \frac{\partial \ell}{\partial u}(I + J_G) = \frac{\partial \ell}{\partial w}. \tag{10}$$

The equation on the right of (10) can be solved for $\partial\ell/\partial u$ using fixed-point iterations since $\mathrm{Lip}(J_G) \leq L < 1$. We now consider $\partial\ell/\partial\theta$. Taking differentiation with respect to $\theta$ while keeping $u$ constant yields

$$\frac{\partial w}{\partial \theta} + \left( J_G \frac{\partial w}{\partial \theta} + \left. \frac{\partial G(x, \theta)}{\partial \theta} \right|_{x=w,\,\theta=\theta} \right) - 0 = 0.$$

Hence, we deduce that

$$\frac{\partial \ell}{\partial \theta} = (-1)\left( \frac{\partial \ell}{\partial w}(I + J_G)^{-1} \right) \left. \frac{\partial G(x, \theta)}{\partial \theta} \right|_{x=w,\,\theta=\theta}. \tag{11}$$

Since the term in parentheses is the same as $\partial\ell/\partial u$, we can reuse the result from the fixed-point iteration for (10). Notice that (11) is similar to the form of

$$\frac{\partial \ell}{\partial \theta} = \frac{\partial \ell}{\partial G(\mathrm{stop\_gradient}(w), \theta)} \frac{\partial G(\mathrm{stop\_gradient}(w), \theta)}{\partial \theta}.$$

Hence, we can backpropagate through $G$ using the standard backpropagation approach by setting the output gradient as $\partial\ell/\partial u$.

## A.6 Proof of Theorem 4

*Proof.* We start by noting that the functions in $\mathcal{R}_L$ are continuously differentiable by construction. In fact, the functions in $\mathcal{I}_L$ are continuously differentiable by the inverse function theorem since the Jacobian $I + J_G$ of $\mathrm{Id} + G$ is nonsingular. Moreover, the functions in $\mathcal{M}_L$ are continuously differentiable since they can be written in terms of the functions of $\mathcal{I}_L$. When the function $F$ such that $F : \mathbb{R}^n \to \mathbb{R}^n$ is invertible, the inverse function theorem implies that $F$ and $F^{-1}$ share the same differentiability, and so do $\mathcal{G}_L, \mathcal{R}_L, \mathcal{I}_L,$ and $\mathcal{M}_L$. Hence, it suffices to consider the Lipschitz

condition. For $x_1, x_2 \in \mathbb{R}^n$, write $p = x_1 - x_2$ and $q = F(x_1) - F(x_2)$. Then, we find that

$$F \in \mathcal{R}_L \Leftrightarrow F \in C^2(\mathbb{R}^n, \mathbb{R}^n), \forall x_1, x_2 \in \mathbb{R}^n \ \|q - p\|_2 \leq L\|p\|_2,$$

$$F \in \mathcal{I}_L \Leftrightarrow F \in C^2(\mathbb{R}^n, \mathbb{R}^n), \forall x_1, x_2 \in \mathbb{R}^n \ \|p - q\|_2 \leq L\|q\|_2$$

$$\overset{(*)}{\Leftrightarrow} F \in C^2(\mathbb{R}^n, \mathbb{R}^n), \forall x_1, x_2 \in \mathbb{R}^n \ \|(1 - L^2)q - p\|_2 \leq L\|p\|_2$$

$$\Leftrightarrow (1 - L^2)F \in \mathcal{R}_L$$

$$\Leftrightarrow F \in \frac{1}{1 - L^2}\mathcal{R}_L,$$

$$F \in \mathcal{M}_L \Leftrightarrow F \in C^2(\mathbb{R}^n, \mathbb{R}^n), \forall x_1, x_2 \in \mathbb{R}^n \ \|q - p\|_2 \leq L\|q + p\|_2$$

$$\overset{(**)}{\Leftrightarrow} F \in C^2(\mathbb{R}^n, \mathbb{R}^n), \forall x_1, x_2 \in \mathbb{R}^n \ \left\|\left(\frac{1 - L^2}{1 + L^2}\right)q - p\right\|_2 \leq \left(\frac{2L}{1 + L^2}\right)\|p\|_2$$

$$\Leftrightarrow \left(\frac{1 - L^2}{1 + L^2}\right)F \in \mathcal{R}_{\frac{2L}{1 + L^2}}$$

$$\Leftrightarrow F \in \left(\frac{1 + L^2}{1 - L^2}\right)\mathcal{R}_{\frac{2L}{1 + L^2}}.$$

Note that the equivalence (*) can be derived as follows:

$$\|p - q\|_2 \leq L\|q\|_2$$

$$\Leftrightarrow \|p - q\|_2^2 \leq L^2\|q\|_2^2$$

$$\Leftrightarrow \|p\|_2^2 - 2\langle p, q\rangle + (1 - L^2)\|q\|_2^2 \leq 0$$

$$\Leftrightarrow (1 - L^2)^2\|q\|_2^2 - 2(1 - L^2)\langle p, q\rangle + (1 - L^2)\|p\|_2^2 \leq 0$$

$$\Leftrightarrow \|(1 - L^2)q - p\|_2^2 \leq L^2\|p\|_2^2$$

$$\Leftrightarrow \|(1 - L^2)q - p\|_2 \leq L\|p\|_2.$$

Also, the equivalence (**) can be calculated as follows:

$$\|q - p\|_2 \leq L\|p + q\|_2$$

$$\Leftrightarrow \|q - p\|_2^2 \leq L^2\|p + q\|_2^2$$

$$\Leftrightarrow (1 - L^2)\|p\|_2^2 - 2(1 + L^2)\langle p, q\rangle + (1 - L^2)\|q\|_2^2 \leq 0$$

$$\Leftrightarrow (1 - L^2)^2\|q\|_2^2 - 2(1 - L^2)(1 + L^2)\langle p, q\rangle + (1 - L^2)^2\|p\|_2^2 \leq 0$$

$$\Leftrightarrow \|(1 - L^2)q - (1 + L^2)p\|_2^2 \leq 4L^2\|p\|_2^2$$

$$\Leftrightarrow \left\|\frac{1 - L^2}{1 + L^2}q - p\right\|_2 \leq \frac{2L}{1 + L^2}\|p\|_2.$$

Hence, the statements (i) and (ii) are proved.

Since $\mathcal{I}_L$ and $\mathcal{M}_L$ are explicitly characterized in terms of $\mathcal{R}_L$ (or $\mathcal{R}_{\frac{2L}{1 + L^2}}$), we utilize this fact to show the statements (iii) and (iv). For (iii), suppose $F \in \mathcal{R}_L$. Then, $F = \mathrm{Id} + G$ for some $G \in \mathcal{G}_L$. Since

$$F = \mathrm{Id} + G = \frac{1 + L^2}{1 - L^2}\left[\mathrm{Id} + \left(\frac{-2L^2}{1 + L^2}\mathrm{Id} + \frac{1 - L^2}{1 + L^2}G\right)\right]$$

and

$$\mathrm{Lip}\left(\frac{-2L^2}{1 + L^2}\mathrm{Id} + \frac{1 - L^2}{1 + L^2}G\right) \leq \frac{2L^2}{1 + L^2} + \frac{1 - L^2}{1 + L^2}L$$

$$= \frac{2L}{1 + L^2} \cdot \left(\frac{1 + 2L - L^2}{2}\right)$$

$$< \text{(since } 0 \leq L < 1)$$

$$< \frac{2L}{1 + L^2},$$

we then have $F \in \mathcal{M}_L$.

For (iv), suppose $F \in \mathcal{I}_L$. Then, $F = \frac{1}{1-L^2}(\mathrm{Id} + G)$ for some $G \in \mathcal{G}_L$. Since

$$F = \frac{1}{1-L^2}(\mathrm{Id} + G) = \frac{1+L^2}{1-L^2}\left[\mathrm{Id} + \left(\frac{-L^2}{1+L^2}\mathrm{Id} + \frac{1}{1+L^2}G\right)\right] \tag{12}$$

and

$$\mathrm{Lip}\left(\frac{-L^2}{1+L^2}\mathrm{Id} + \frac{1}{1+L^2}G\right) \leq \frac{L^2}{1+L^2} + \frac{1}{1+L^2}L = \frac{2L}{1+L^2}\cdot\left(\frac{1+L}{2}\right) < \frac{2L}{1+L^2},$$

we obtain that $F \in \mathcal{M}_L$. We finally notice that the above derivation, indeed, shows that the Lipschitz constant of the residual part $\mathrm{Lip}\left(\frac{1-L^2}{1+L^2}F - \mathrm{Id}\right)$ is always *smaller* than $\frac{2L}{1+L^2}$ for functions $F$ in $\mathcal{R}_L$ or $\mathcal{I}_L$. Hence, both $\mathcal{R}_L$ and $\mathcal{I}_L$ do not include the following functions

$$f_1(x) = \frac{1-L}{1+L}x \quad \text{and} \quad f_2(x) = \frac{1+L}{1-L}x,$$

which are in $\mathcal{M}_L$, since

$$\mathrm{Lip}\left(\frac{1-L^2}{1+L^2}f_1 - \mathrm{Id}\right) = \mathrm{Lip}\left(\frac{1-L^2}{1+L^2}f_2 - \mathrm{Id}\right) = \frac{2L}{1+L^2}.$$

Hence (iii) and (iv) hold. See Figure 3 in the main text for visualization, where $p$ is fixed as a unit vector pointing in the $+y$ direction. ∎

## A.7  Equivalence under the limit $L \to 1^-$

For each function class, if we consider the union of the "subscript $L$-sets" for all $0 \leq L < 1$, then we have the following theorem.

**Theorem 6.** *Define the set $A$ as*

$$A := \left\{F \in C^2(\mathbb{R}^n, \mathbb{R}^n) \,\middle|\, F \text{ is } \eta\text{-strongly monotone and } \nu\text{-Lipschitz for some } \eta, \nu > 0\right\}.$$

*Then, $\mathbb{R}^+\mathcal{R}_{[0,1)} = \mathbb{R}^+\mathcal{I}_{[0,1)} = \mathbb{R}^+\mathcal{M}_{[0,1)} = A$. Here, $\mathcal{R}_{[0,1)} = \cup_{L \in [0,1)}\mathcal{R}_L$, and the same for $\mathcal{I}_{[0,1)}$ and $\mathcal{M}_{[0,1)}$.*

*Proof.* The equivalence between $\mathbb{R}^+\mathcal{R}_{[0,1)}$ and $\mathbb{R}^+\mathcal{I}_{[0,1)}$ directly follows from Theorem 4. The equivalence between $\mathbb{R}^+\mathcal{R}_{[0,1)}$ and $\mathbb{R}^+\mathcal{M}_{[0,1)}$ also follows from Theorem 4 by noting that the function $f(t) = 2t/(1+t^2)$ is a bijection from $[0,1)$ to itself.

It remains to prove the last equality. First, $\mathbb{R}^+\mathcal{M}_{[0,1)} \subseteq A$ holds because each function in $\mathcal{M}_L$ is $(1-L)/(1+L)$-strongly monotone and $(1+L)/(1-L)$-Lipschitz for all $0 \leq L < 1$, by Theorem 7.

We now prove the reverse direction. For any $F \in A$, there exists a $0 < K \leq 1$ such that $K \leq \eta$ and $\nu \leq 1/K$, so that $F$ is $K$-strongly monotone and $1/K$-Lipschitz. For any $x_1, x_2 \in \mathbb{R}^n$, write $p = x_1 - x_2$ and $q = F(x_1) - F(x_2)$ as before. We have

$$\|q\|_2 \leq \frac{1}{K}\|p\|_2, \quad \langle q, p \rangle \geq K\|p\|_2^2. \tag{13}$$

Now, $F \in \mathcal{M}_L$ for some $0 \leq L < 1$ if and only if

$$\|q - p\|_2 \leq L\|q + p\|_2 \tag{14}$$

for all $x_1, x_2 \in \mathbb{R}^n$. To find a sufficient condition for $L$, we manipulate this equation:

$$\|q - p\|_2 \leq L\|q + p\|_2$$
$$\Leftrightarrow 2(L^2 + 1)\langle q, p \rangle - (1 - L^2)(\|p\|_2^2 + \|q\|_2^2) \geq 0.$$

Since

$$2(L^2 + 1)\langle q, p \rangle - (1 - L^2)(\|p\|_2^2 + \|q\|_2^2) \geq \left[2(L^2 + 1)K - (1 - L^2)\left(1 + \frac{1}{K^2}\right)\right]\|p\|_2^2,$$

by (13), the inequality (14) will hold if we can find a $0 \leq L < 1$ such that

$$2(L^2 + 1)K - (1 - L^2)\left(1 + \frac{1}{K^2}\right) \geq 0,$$

which is indeed solved by any choice of $L$ satisfying

$$\sqrt{\frac{1 + K^2 - 2K^3}{1 + K^2 + 2K^3}} \leq L < 1.$$

Since

$$\sqrt{\frac{1 + K^2 - 2K^3}{1 + K^2 + 2K^3}} < 1,$$

for all $0 < K \leq 1$, there always exists such an $L$. This proves $A \subseteq \mathbb{R}^+\mathcal{M}_{[0,1)}$. Hence, $\mathbb{R}^+\mathcal{M}_{[0,1)} = A$. ∎

### A.8    The properties of monotone operators built from $L$-Lipschitz operators

**Theorem 7.** *Let $G$ be an $L$-Lipschitz operator for $L < 1$. The monotone operator $F$ having $G$ as its Cayley operator is (i) $\eta$-strongly monotone and (ii) $\nu$-Lipschitz for*

$$\eta = \frac{1 - L}{1 + L} \quad and \quad \nu = \frac{1 + L}{1 - L}.$$

*Proof.* Let $(x_1, y_1)$, $(x_2, y_2) \in F$ and $a_i = x_i + y_i$, $b_i = x_i - y_i$ for $i = 1, 2$. We first note that $\|b_1 - b_2\|_2 = \|G(a_1) - G(a_2)\|_2 \leq L\|a_1 - a_2\|_2$ by the definition of Lipschitz continuity.

(i) Let's prove the $\eta$-strongly monotonicity. We have

$$\langle y_1 - y_2, x_1 - x_2 \rangle = \left\langle \frac{1}{2}(a_1 - b_1) - \frac{1}{2}(a_2 - b_2), \frac{1}{2}(a_1 + b_1) - \frac{1}{2}(a_2 + b_2) \right\rangle$$

$$= \frac{1}{4}\left(\|a_1 - a_2\|_2^2 - \|b_1 - b_2\|_2^2\right) \geq \frac{1 - L^2}{4}\|a_1 - a_2\|_2^2,$$

and

$$\|x_1 - x_2\|_2 = \left\|\frac{1}{2}(a_1 + b_1) - \frac{1}{2}(a_2 + b_2)\right\|_2$$

$$\leq \frac{1}{2}\left(\|a_1 - a_2\|_2 + \|b_1 - b_2\|_2\right) \leq \frac{1 + L}{2}\|a_1 - a_2\|_2.$$

Combining the two inequalities we have

$$\langle y_1 - y_2, x_1 - x_2 \rangle \geq \frac{1}{4}(1 - L^2)\left(\frac{2}{1 + L}\|x_1 - x_2\|_2\right)^2 = \frac{1 - L}{1 + L}\|x_1 - x_2\|_2^2,$$

which implies the $\eta$-strong monotonicity. Equality holds with $G(x) = Lx$.

(ii) Let's prove the $\nu$-Lipschitzness. We have

$$\|y_1 - y_2\|_2 = \left\|\frac{1}{2}(a_1 - b_1) - \frac{1}{2}(a_2 - b_2)\right\|_2$$

$$\leq \frac{1}{2}\left(\|a_1 - a_2\|_2 + \|b_1 - b_2\|_2\right) \leq \frac{1 + L}{2}\|a_1 - a_2\|_2,$$

and

$$\|x_1 - x_2\|_2 = \left\|\frac{1}{2}(a_1 + b_1) - \frac{1}{2}(a_2 + b_2)\right\|_2$$

$$\geq \frac{1}{2}\left(\|a_1 - a_2\|_2 - \|b_1 - b_2\|_2\right) \geq \frac{1 - L}{2}\|a_1 - a_2\|_2.$$

Combining the two inequalities we have

$$\|y_1 - y_2\|_2 \leq \frac{1 + L}{2}\|a_1 - a_2\|_2 \leq \frac{1 + L}{1 - L}\|x_1 - x_2\|_2,$$

which implies the $\nu$-Lipschitzness. Equality holds with $G(x) = -Lx$. ∎

# B  Computation

## B.1  Forward and backward algorithms for a single monotone layer

We describe the forward and backward algorithms for a single monotone layer. Both utilize fixed-point iterations, but the forward pass uses non-linear functions, whereas the backward pass uses linear functions. We let $G_\theta$ denote the $G$-network in Definition 2 with parameters $\theta$, $F_\theta$ the monotone formulation of $G_\theta$, and $\ell$ the loss function of the whole model. Note that in Algorithm 3, we maintain the values of $\alpha$ and $\beta$ for each sample in minibatch. Also note that in Algorithm 1, the maximum number of iterations $n_{\max}$ applies separately to the first and the second fixed-point algorithm. We choose $(\epsilon, n_{\max}) = (10^{-6}, 2000)$ for the forward pass and $(\epsilon, n_{\max}) = (10^{-9}, 100)$ for the backward pass.

For time and memory efficiency, we use the Neumann gradient estimator following [6], defined by the following equation. The last expression has the form $\partial\mathcal{L}/\partial\theta$; we use $\mathcal{L}$ as the *surrogate loss* for estimating the gradients with respect to $\theta$.

$$
\begin{aligned}
\frac{\partial}{\partial\theta}\log\det J_{F_\theta} &= \frac{\partial}{\partial\theta}\mathrm{tr}\left[\log(I - J_{G_\theta}) - \log(I + J_{G_\theta})\right] \\
&= \frac{\partial}{\partial\theta}\mathrm{tr}\left[\sum_{k=1}^{\infty}\frac{(-1) - (-1)^{k+1}}{k}J_{G_\theta}^{k}\right] \\
&= \mathrm{tr}\left[\sum_{k=1}^{\infty}((-1) - (-1)^{k+1})\frac{\partial J_{G_\theta}}{\partial\theta}J_{G_\theta}^{k-1}\right] \\
&= \mathbb{E}_{n\sim p_N(n), v\sim\mathcal{N}(0,I)}\left[\sum_{k=1}^{n}\frac{(-1) - (-1)^{k+1}}{P(N \geq k)}v^T\frac{\partial J_{G_\theta}}{\partial\theta}J_{G_\theta}^{k-1}v\right] \\
&= \frac{\partial}{\partial\theta}\mathbb{E}_{n\sim p_N(n), v\sim\mathcal{N}(0,I)}\left[\sum_{k=1}^{n}\frac{(-1) - (-1)^{k+1}}{P(N \geq k)}v^T J_{G_\theta}\mathrm{stop\_gradient}\left(J_{G_\theta}^{k-1}v\right)\right].
\end{aligned}
\tag{15}
$$

---

**Algorithm 1** Forward algorithm for a single monotone layer

---

1: **Require:** $G_\theta$, tolerance $\epsilon$, max. number of iterations $n_{\max}$
2: **Input:** $x$
3: **Output:** $z = F_\theta(x)$, $p = \log\det\left|\dfrac{\partial z}{\partial x}\right|$
4: $u \leftarrow 2x$
5: $w \leftarrow \mathrm{FixedPointSolver}(f(y) = u - G_\theta(y), x, \epsilon, n_{\max})$
6: $z \leftarrow w - x$
7: **if** training **then**
8:     Estimate $p$ using the surrogate loss (15).
9: **else**
10:     Estimate $p$ using the true estimator (6).
11: **end if**
12: Return $z$ and $p$

---

---

**Algorithm 2** Backward algorithm for a single monotone layer

---

1: **Require:** $G_\theta$, tolerance $\epsilon$, max. number of iterations $n_{\max}$

2: **Input:** $x$, $z$, $p$, $\dfrac{\partial \ell}{\partial z}$, $\dfrac{\partial \ell}{\partial p}$

3: **Output:** $\dfrac{\partial \ell}{\partial x}$, $\dfrac{\partial \ell}{\partial \theta}$

4: Backpropagate through $z$ to $w$ and $x$

5: Backpropagate through $w$ to $u$ and $\theta$:

6: $\qquad g \leftarrow \text{FixedPointSolver}\left( f(y) = \dfrac{\partial \ell}{\partial w} - y\dfrac{\partial G_\theta(w)}{\partial w}, \ \dfrac{\partial \ell}{\partial w}, \ \epsilon, \ n_{\max} \right)$

7: $\qquad \dfrac{\partial \ell}{\partial u} \leftarrow g$

8: $\qquad \dfrac{\partial \ell}{\partial \theta} \leftarrow -g\dfrac{\partial G_\theta(w)}{\partial \theta}$

9: Backpropagate through $u$ to $x$

10: Backpropagate through $p$ to $x$ and $\theta$

11: Return the gradients $\dfrac{\partial \ell}{\partial x}$ and $\dfrac{\partial \ell}{\partial \theta}$

---

---

**Algorithm 3** Fixed-point solver

---

1: **Require:** input function (contraction mapping) $f$, tolerance $\epsilon$, max. number of iterations $n_{\max}$

2: **Input:** $y_0$, the starting point of the fixed-point iteration

3: **Output:** $y$, the fixed point of $f$ satisfying $y = f(y)$

4: $y_1 \leftarrow f(y_0)$

5: $f_{\text{prev}} \leftarrow y_1$

6: $dy_{\text{prev}} \leftarrow y_1 - y_0$

7: $y_{\text{curr}} \leftarrow y_1$

8: **while** error higher than the tolerance $\epsilon$ and iteration limit $n_{\max}$ not reached **do**

9: $\qquad$ **if** error has not improved in 10 recent iterations **then**

10: $\qquad\qquad$ **break**

11: $\qquad$ **end if**

12: $\qquad f_{\text{curr}} \leftarrow f(y_{\text{curr}})$

13: $\qquad dy_{\text{curr}} \leftarrow f_{\text{curr}} - y_{\text{curr}}$

14: $\qquad d^2 y_{\text{curr}} \leftarrow dy_{\text{curr}} - dy_{\text{prev}}$

15: $\qquad \beta \leftarrow \dfrac{\langle d^2 y_{\text{curr}}, dy_{\text{curr}} \rangle}{||d^2 y_{\text{curr}}||_2^2 + 10^{-8}}$

16: $\qquad y_{\text{next}} \leftarrow f_{\text{curr}} - \beta(f_{\text{curr}} - f_{\text{prev}})$

17: $\qquad dy_{\text{prev}} \leftarrow dy_{\text{curr}}$

18: $\qquad f_{\text{prev}} \leftarrow f_{\text{curr}}$

19: $\qquad y_{\text{curr}} \leftarrow y_{\text{next}}$

20: **end while**

21: **if** error within the tolerance $\epsilon$ **then**

22: $\qquad$ Return $y_{\text{curr}}$

23: **end if**

24: $\alpha \leftarrow 0.5$

25: **while** error higher than the tolerance $\epsilon$ and iteration limit $n_{\max}$ not reached **do**

26: $\qquad$ **if** error has not improved in recent 30 iterations **then**

27: $\qquad\qquad \alpha \leftarrow \max\{0.9\alpha, 0.1\}$

28: $\qquad\qquad$ Reset the counter

29: $\qquad$ **end if**

30: $\qquad y_{\text{curr}} \leftarrow (1 - \alpha)y_{\text{curr}} + \alpha f(y_{\text{curr}})$

31: **end while**

32: Print the current error for logging purposes

33: Return $y_{\text{curr}}$

---

## C Experimental details

All of our experiments are implemented with PyTorch, based on the public code of i-DenseNets [7]. We have implemented CPila as a C++ CUDA extension for speedup. In all experiments, we disable the TensorFloat32 (TF32) and the CUDA benchmarks option to prevent the non-deterministic computation from affecting the convergence of the fixed-point iterations.

### C.1 1D toy experiment

**Training data.** We fit a strongly monotone function $s : \mathbb{R} \to \mathbb{R}$ defined as follows:

$$s(x) = t(x+2) + t(x+1) + t(x) + t(x-1), \tag{16}$$

where $t : \mathbb{R} \to \mathbb{R}$ is defined as

$$t(x) = \begin{cases} 0, & x < 0 \\ \max\left\{\alpha x, \frac{1}{\alpha}(x - 1 + \alpha)\right\}, & 0 \leq x \leq 1 \\ 1, & x > 1 \end{cases} \tag{17}$$

with the choice of $\alpha = 0.05$. The function $s$ is considered on the interval $[-2, 2]$ only.

**Training objective.** We use the mean squared error (MSE) for the loss function and report the best test MSE in Figure 4.

**Training procedure.** We use Adam [42] for optimization with $(\beta_1, \beta_2) = (0.9, 0.99)$, eps $= 10^{-8}$, and an initial learning rate of 0.01, which drops to 0.002 and 0.0004 after 5,000 and 10,000 iterations, respectively. We train for 15,000 iterations in total. We do not apply weight decay. We enable the learnable concatenation at the beginning. For training, we randomly sample 5,000 points uniformly from $[-2, 2]$ on each iteration; for testing, we use 20,001 uniformly spaced points on $[-2, 2]$. We run tests every 100 iterations. We train each of the four models using a single NVIDIA RTX 2080 Ti.

**Model architecture.** There are four building blocks: ResidualBlock (RB), InverseResidualBlock (IRB), MonotoneBlock (MB), and ActNorm (learnable scaling; LS). RB, IRB, and MB share the same $G$-network built by four DenseNet concatenations, each adding 128 channels, followed by a final linear layer. We use Concatenated ReLU (CReLU) as the activation function. The Lipschitz constant for spectral normalization and dense layers are 0.99 and 0.99, respectively; the tolerance for spectral normalization is $10^{-4}$. ActNorm [4] learns a positive scaling parameter and a bias for each channel. It acts as a learnable scaling (LS) layer that applies to the whole input since there is only one channel for 1D toy experiments. The architectures for each model are as follows:

- (a) 2 RBs w/o LS:
$$\text{ResidualBlock} \to \text{ResidualBlock}$$
- (b) 2 RBs w/ LS:
$$\text{ActNorm} \to \text{ResidualBlock} \to \text{ActNorm} \to \text{ResidualBlock} \to \text{ActNorm}$$
- (c) 1 RB + 1 IRB w/ LS:
$$\text{ActNorm} \to \text{ResidualBlock} \to \text{ActNorm} \to \text{InverseResidualBlock} \to \text{ActNorm}$$
- (d) 2 MBs w/ LS:
$$\text{ActNorm} \to \text{MonotoneBlock} \to \text{ActNorm} \to \text{MonotoneBlock} \to \text{ActNorm}$$

### C.2 2D toy experiments

**Training data.** We use the eight 2D toy densities provided with the source code of i-DenseNets [7], which originate from [36] to the best of our knowledge. They consist of '2 Spirals', '8 Gaussians', 'Checkerboard', 'Circles', 'Moons', 'Pinwheel', 'Rings', and 'Swissroll'.

**Training objective.** We use the typical log-likelihood objective of normalizing flows. Note that we evaluate the Jacobian determinant exactly using the determinant formula for 2D matrices since the low dimensionality (2D) allows for the fast and exact computation of the Jacobian determinant without needing to resort to the stochastic estimation.

**Training procedure.** We use Adam [42] for optimization with $(\beta_1, \beta_2) = (0.9, 0.99)$, eps $= 10^{-8}$, and the learning rate of 0.001. We train for 50,000 iterations in total. We apply the weight decay of

Table 4: The training setup for image density estimation tasks. LC: learnable concatenation.

|  | MNIST | CIFAR-10 | ImageNet32 | ImageNet64 |
|---|---|---|---|---|
| Learning rate | 0.001 | 0.001 | 0.004 | 0.004 |
| Batch size | 64 | 64 | 256 | 256 |
| Number of epochs | 100 | 1,000 | 20 | 20 |
| Enable LC | After 25 epochs | After 25 epochs | At beginning | At beginning |

$10^{-5}$. We enable the learnable concatenation after 25,000 iterations. For training, we sample 500 points from the target distribution on each iteration; for testing, we sample 10,000 points. We run tests every 100 iterations. We train i-DenseNets and Monotone Flows for each dataset using a single NVIDIA RTX 2080 Ti.

**Model architecture.** The $G$-networks for i-DenseNets and Monotone Flows only differ in the choice of the activation function, where we use CLipSwish and CPila, respectively. Otherwise, they share the same structure, built by three DenseNet concatenations, each adding 16 channels, followed by a final linear layer. The Lipschitz constant for spectral normalization and dense layers are 0.90 and 0.98, respectively. We do not explicitly specify the tolerance for spectral normalization; instead, we run five iterations on each spectral normalization update regardless of the current error. We do not use ActNorm layers for 2D toy experiments. The architectures for each model are as follows; we use ten blocks following the 2D toy experiment setup of [7].

- i-DenseNets: $10 \times$ [ResidualBlock]
- Monotone Flows: $10 \times$ [MonotoneBlock]

### C.3 Image experiments

**Training data.** We use MNIST, CIFAR-10, ImageNet32, and ImageNet64. For CIFAR-10, we apply a random horizontal flip with a probability of 0.5 and do nothing for the other three datasets. We then add noise for uniform dequantization. The images keep their original sizes, which are $28 \times 28$, $32 \times 32$, $32 \times 32$, and $64 \times 64$, respectively.

**Training objective.** We use the typical log-likelihood objective of normalizing flows. We evaluate the Jacobian determinant using the stochastic estimator, with ten exact terms plus additional terms with the Russian roulette estimator. In contrast to the 2D toy experiments, we use the bits per dimension, defined as the $\log_2$-likelihood divided by the input dimension, with the compensation for the quantization levels. Since there are 256 quantization levels for the image data, we add 8 to the $\log_2$-likelihood.

**Training procedure.** We use Adam [42] for optimization with $(\beta_1, \beta_2) = (0.9, 0.99)$ and eps = $10^{-8}$. We do not apply weight decay. Other options vary among the four datasets, as described in Table 4. We run tests at the end of every epoch. We use four RTX 3090 GPUs for MNIST and CIFAR-10 and eight A100 GPUs for ImageNet32 and ImageNet64.

**Model architecture.** We use monotone blocks for each model, with the $G$-networks built by three DenseNet concatenations, each adding 172 channels, followed by a final linear layer. We use Concatenated Pila (CPila) as the activation function. The Lipschitz constant for spectral normalization and dense layers are 0.98 and 0.98, respectively. We set the tolerance for spectral normalization to $10^{-3}$. To implement a multi-scale architecture, we utilize invertible downsampling operations ('Squeeze'), which package the pixels of two-dimensional $2 \times 2$ cells into a 4-dimensional vector, hence converting an input of the shape $C \times H \times W$ into the shape $4C \times \frac{H}{2} \times \frac{W}{2}$. Layers with no invertible downscaling layers in between share the same scale. We use four fully-connected layers at the end of the models with a DenseNet depth and growth of 3 and 16. Note that the fully-connected layers first project the input into a vector of 64 dimensions, apply the DenseNet formulation, and then project back to the original dimensions. 'FactorOut' is the operation that factors out half of the variables at the transition of one scale to another scale, splitting across channels. 'Squeeze2d' and 'FactorOut' applied in succession converts an input of the shape $C \times H \times W$ into the shape $2C \times \frac{H}{2} \times \frac{W}{2}$. The architectural choices that differ across the datasets are displayed in Table 5. The architectures for each model are as follows:

Table 5: The model setup for image density estimation tasks.

|  | MNIST | CIFAR-10 | ImageNet32 | ImageNet64 |
|---|---|---|---|---|
| Logit transform's $\alpha$ | $10^{-6}$ | 0.05 | 0.05 | 0.05 |
| Number of scales | 3 | 3 | 3 | 3 |
| Number of flow blocks per scale | 16 | 16 | 32 | 32 |
| Factor out at the end of each scale | No | No | No | Yes |

The architectures for each model are as follows. Conv: convolutional layers; FC: fully connected layers.

- MNIST: LogitTransform($10^{-6}$) $\rightarrow$ ActNorm2d $\rightarrow$ 16 $\times$ [MonotoneBlock$_{\text{Conv}}$ $\rightarrow$ ActNorm2d] $\rightarrow$ Squeeze2d $\rightarrow$ 16 $\times$ [MonotoneBlock$_{\text{Conv}}$ $\rightarrow$ ActNorm2d] $\rightarrow$ Squeeze2d $\rightarrow$ 15 $\times$ [MonotoneBlock$_{\text{Conv}}$ $\rightarrow$ ActNorm2d] $\rightarrow$ MonotoneBlock$_{\text{Conv}}$ $\rightarrow$ ActNorm1d $\rightarrow$ 4 $\times$ [MonotoneBlock$_{\text{FC}}$ $\rightarrow$ ActNorm1d]

- CIFAR-10: LogitTransform(0.05) $\rightarrow$ ActNorm2d $\rightarrow$ 16 $\times$ [MonotoneBlock$_{\text{Conv}}$ $\rightarrow$ ActNorm2d] $\rightarrow$ Squeeze2d $\rightarrow$ 16 $\times$ [MonotoneBlock$_{\text{Conv}}$ $\rightarrow$ ActNorm2d] $\rightarrow$ Squeeze2d $\rightarrow$ 15 $\times$ [MonotoneBlock$_{\text{Conv}}$ $\rightarrow$ ActNorm2d] $\rightarrow$ MonotoneBlock$_{\text{Conv}}$ $\rightarrow$ ActNorm1d $\rightarrow$ 4 $\times$ [MonotoneBlock$_{\text{FC}}$ $\rightarrow$ ActNorm1d]

- ImageNet32: LogitTransform(0.05) $\rightarrow$ ActNorm2d $\rightarrow$ 32 $\times$ [MonotoneBlock$_{\text{Conv}}$ $\rightarrow$ ActNorm2d] $\rightarrow$ Squeeze2d $\rightarrow$ 32 $\times$ [MonotoneBlock$_{\text{Conv}}$ $\rightarrow$ ActNorm2d] $\rightarrow$ Squeeze2d $\rightarrow$ 31 $\times$ [MonotoneBlock$_{\text{Conv}}$ $\rightarrow$ ActNorm2d] $\rightarrow$ MonotoneBlock$_{\text{Conv}}$ $\rightarrow$ ActNorm1d $\rightarrow$ 4 $\times$ [MonotoneBlock$_{\text{FC}}$ $\rightarrow$ ActNorm1d]

- ImageNet64: LogitTransform(0.05) $\rightarrow$ ActNorm2d $\rightarrow$ 32 $\times$ [MonotoneBlock$_{\text{Conv}}$ $\rightarrow$ ActNorm2d] $\rightarrow$ Squeeze2d $\rightarrow$ FactorOut $\rightarrow$ 32 $\times$ [MonotoneBlock$_{\text{Conv}}$ $\rightarrow$ ActNorm2d] $\rightarrow$ Squeeze2d $\rightarrow$ FactorOut $\rightarrow$ 31 $\times$ [MonotoneBlock$_{\text{Conv}}$ $\rightarrow$ ActNorm2d] $\rightarrow$ MonotoneBlock$_{\text{Conv}}$ $\rightarrow$ ActNorm1d $\rightarrow$ 4 $\times$ [MonotoneBlock$_{\text{FC}}$ $\rightarrow$ ActNorm1d]

## C.4 Variational dequantization

The experiments with variational dequantization use the same main network structure used for the experiments with uniform dequantization. The variational dequantization network, now added at the beginning of the network, uses three invertible DenseNet blocks, which uses CPila activation and does not use monotone formulation. Conditioning is performed by injecting conditional feature vectors into each step inside each invertible DenseNet block. The conditional feature vectors are computed from the input image using ResNet-based neural networks.

## C.5 Ablation studies

The ablation studies are mostly the same as the image experiments discussed in Appendix C.3. Thus, we only highlight the differences:

- When ablating the monotone formulation, we replace each MonotoneBlock$_{\text{Conv/FC}}$ with ResidualBlock$_{\text{Conv/FC}}$, while keeping the $G$-networks the same.

- When ablating the CPila activation function, we replace all occurrences of CPila in each MonotoneBlock$_{\text{Conv/FC}}$ or ResidualBlock$_{\text{Conv/FC}}$ with CLipSwish.

## C.6 Classification experiments

To verify the capacity of Monotone Flows, we perform classification experiments with the same network used for image density estimation tasks.

**Training data.** We use the CIFAR-10 dataset. We apply the standard data augmentation for CIFAR-10, which amounts to padding the input image by 4 pixels, randomly cropping the image back to the original size, and applying a random horizontal flip. Then, we add noise following the image density estimation experiments.

Table 6: Classification results on CIFAR-10.

| | i-DenseNet | Monotone Flow |
|---|---|---|
| Tiny $(k = 1)$ | 86.7 % | **88.9 %** |
| Small $(k = 4)$ | 90.2 % | **91.8 %** |
| Large $(k = 16)$ | 92.5 % | **93.4 %** |

**Training objective.** We use the cross-entropy loss during training and report the average test accuracy over the last five epochs.

**Training procedure.** We use Adam [42] for optimization with $(\beta_1, \beta_2) = (0.9, 0.99)$, eps $= 10^{-8}$, and the learning rate of 0.001. We do not apply weight decay. We enable the learnable concatenation after 25 epochs. We train for 200 epochs with a batch size of 128 and run tests at the end of every epoch. We train each model using two NVIDIA V100 GPUs.

**Model architecture.** Each model has the following structure: Mean-Std normalization $\rightarrow k \times$ [MonotoneBlock$_{\text{Conv}}$ $\rightarrow$ ActNorm2d] $\rightarrow$ Squeeze2d $\rightarrow k \times$ [MonotoneBlock$_{\text{Conv}}$ $\rightarrow$ ActNorm2d] $\rightarrow$ Squeeze2d $\rightarrow k \times$ [MonotoneBlock$_{\text{Conv}}$ $\rightarrow$ ActNorm2d]. The $G$-network has a DenseNet depth and growth of 3 and 80. There are no fully-connected layers at the end. We use $k = 1$ for tiny models, $k = 4$ for small models, and $k = 16$ for large models. For i-DenseNets, each MonotoneBlock gets replaced with a ResidualBlock with the CLipSwish activation function. Classification heads are attached after each Squeeze2d and at the end of the model. Each head consists of Conv2d $\rightarrow$ ActNorm2d $\rightarrow$ ReLU $\rightarrow$ AvgPool2d, yielding a 256-dimensional feature vector per head. The vectors are concatenated into a 768-dimensional vector, passed through a linear layer, and then fed into a softmax layer for classification.

**Classification results.** We present the results in Table 6. The results clearly demonstrate that Monotone Flows consistently outperform i-DenseNets for all model sizes considered.

## C.7 Training curve for CIFAR-10 density estimation

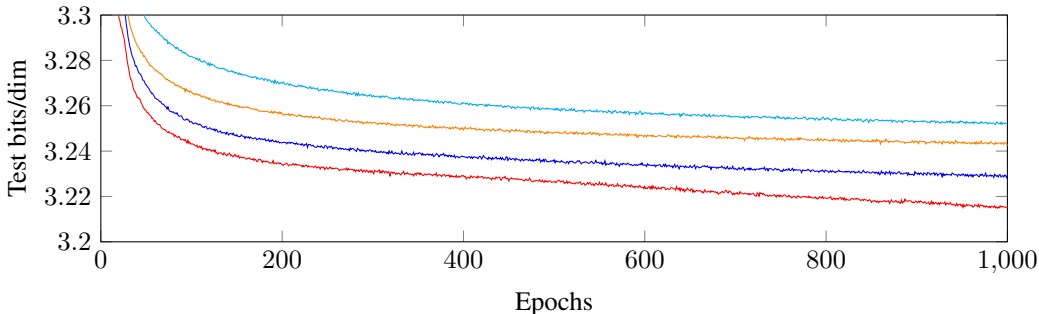

Figure 7: Loss curves for CIFAR-10 training with uniform dequantization. From top to bottom, the cyan curve denotes the baseline i-DenseNets model; the orange curve denotes our model with monotone formulation ablated; the blue curve denotes our model with CPila ablated; the red curve denotes our full model.

## D  Full toy results

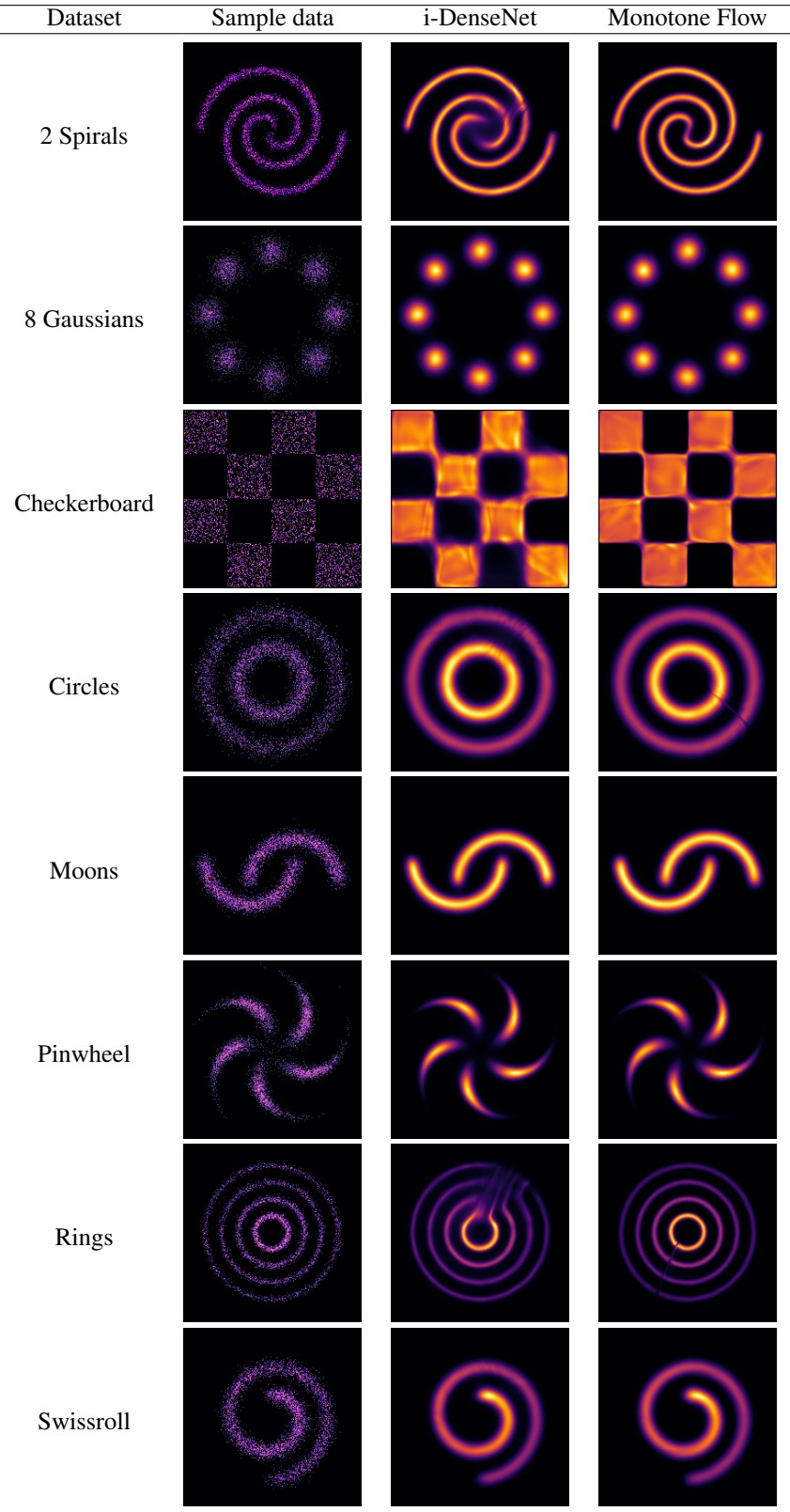

Figure 8: Full toy results.

# E Image samples

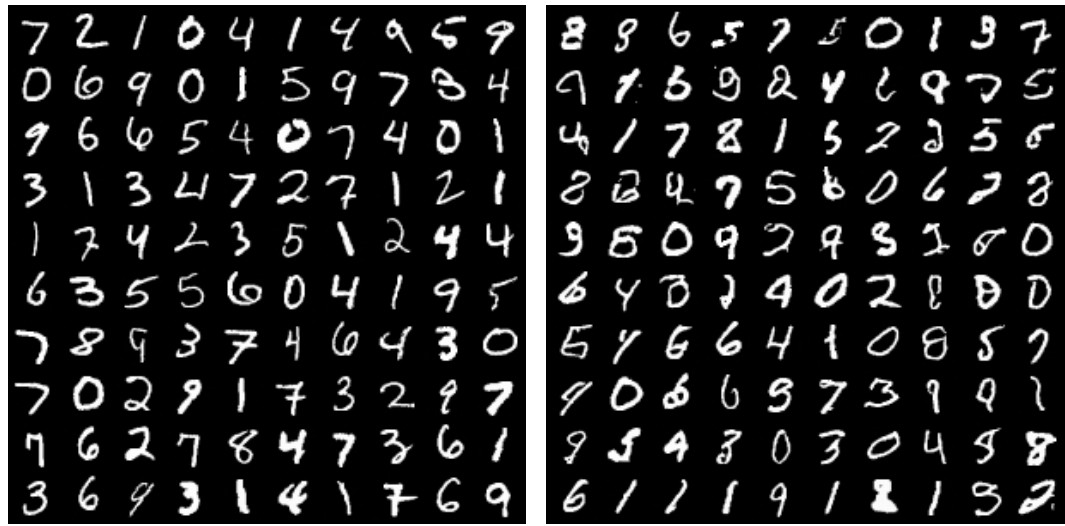

(a) MNIST train data.

(b) Monotone Flows trained on MNIST.

Figure 9: Train data and generated samples of MNIST.

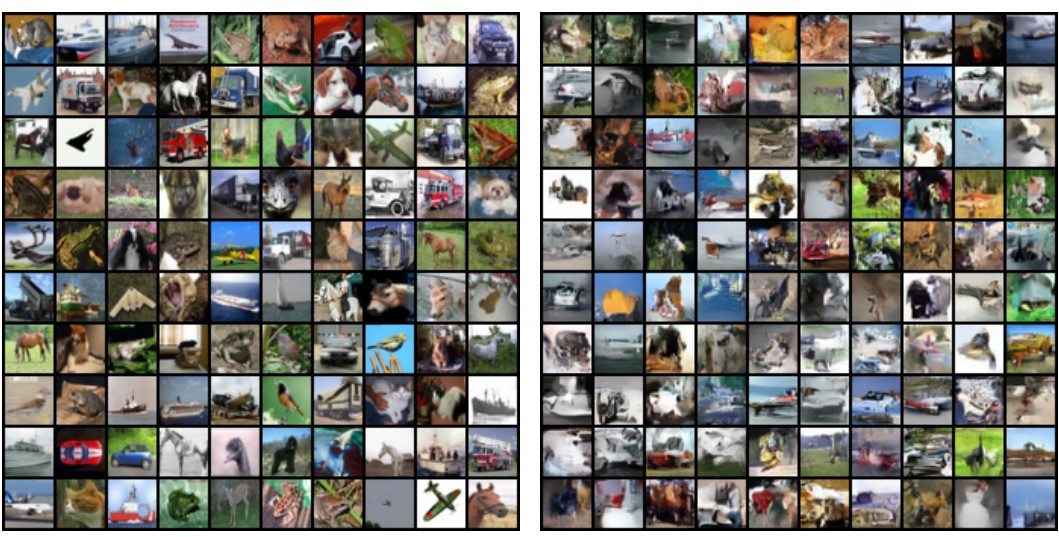

(a) CIFAR-10 train data.

(b) Monotone Flows trained on CIFAR-10.

Figure 10: Train data and generated samples of CIFAR-10.

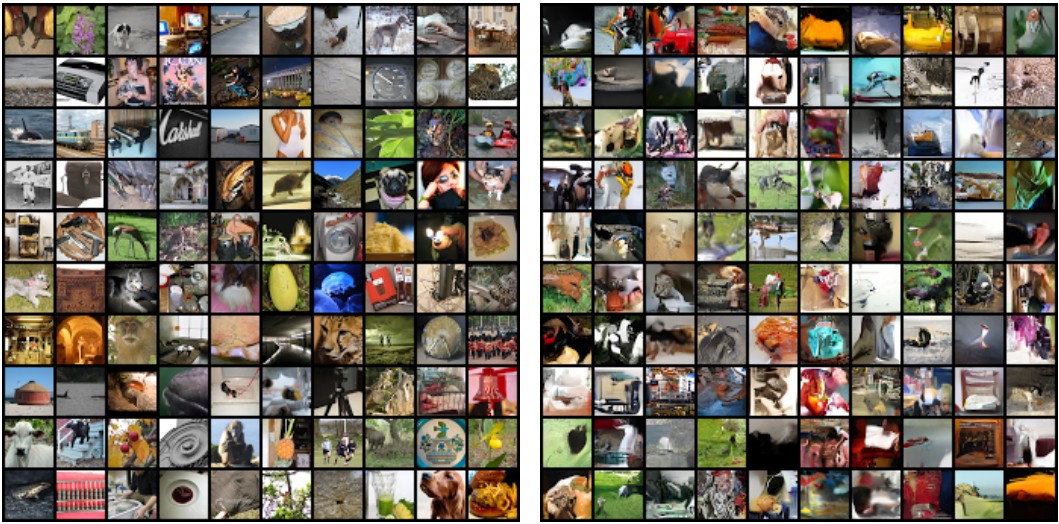

(a) ImageNet32 train data.

(b) Monotone Flows trained on ImageNet32.

Figure 11: Train data and generated samples of ImageNet32.

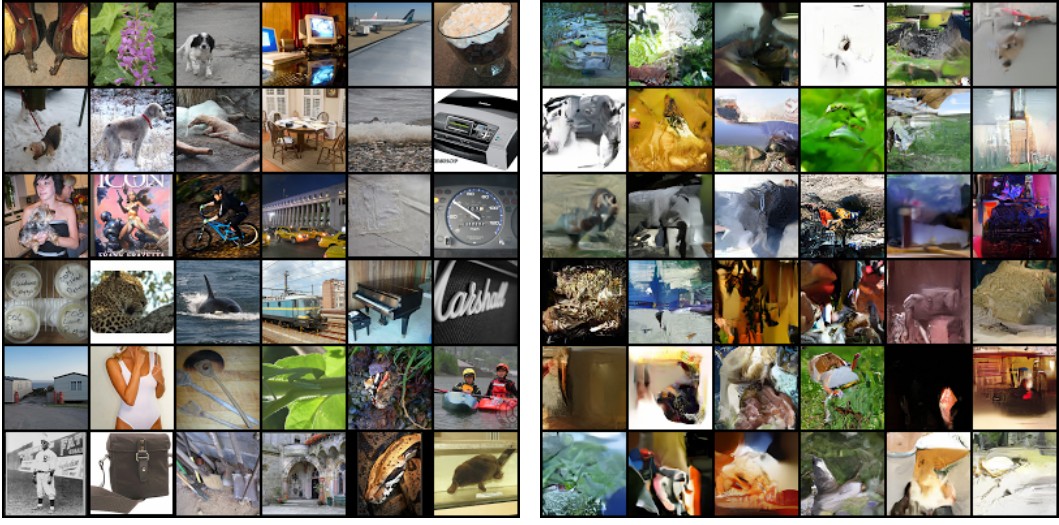

(a) ImageNet64 train data.

(b) Monotone Flows trained on ImageNet64.

Figure 12: Train data and generated samples of ImageNet64.

# F   Limitations and negative societal impact

Similar to previous works: i-ResNets, Residual Flows, i-DenseNets, and Implicit Normalizing Flows, our Monotone Flows involve fixed-point equations, which often leads to computational overhead. However, the speed will likely improve as the methods for solving fixed-point equations, including neural solvers and better initialization schemes, continue to evolve. Also, although our model as a normalizing flow has the advantage of training stability and not suffering from mode collapses, the generated images generally do not yet achieve high fidelity. This is a common weakness of normalizing flow models, and we leave this for future work.

For potential negative societal impact, we note that while the improved modeling capacity of Monotone Flows can benefit many downstream applications of normalizing flows, they have the risk of being misused for the generation of fake images, just like other generative models. Hence, they may facilitate the spread of misinformation or deep fakes, negatively impacting society.