# OpenReview forum: "Invertible Monotone Operators for Normalizing Flows"
_NeurIPS.cc/2022/Conference — NeurIPS 2022 Accept_

### Official Review · Reviewer_crL7 · 2022-07-02

**Rating:** 7
**Confidence:** 3
**Soundness:** 3 good
**Presentation:** 3 good
**Contribution:** 3 good

**Summary:**

The paper proposed a bigger class of invertible residual-type blocks based on monotone operators. Theoretically, the paper shows with the same Lipschitz constraints on the network, their blocks can express more functions than the simple residual formulation. They also provide algorithms to estimate log-det and compute gradients. They then introduced Pila activation to tackle vanishing gradients. Experimentally, they show the proposed flow achieves better density estimation results than other flow-based methods on some standard image datasets.

**Questions:**

- Thm 4 says $\mathcal{M}_L=\frac{1+L^2}{1-L^2} \mathcal{R}_K, K=\frac{2L}{1+L^2}$. Since $K<1$ when $L<1$, is choosing a function from $\mathcal{G}_L$ and applying your formulation equivalent to choosing a function from $\mathcal{G}_K$ and applying the residual formulation, if we disregard the constant $\frac{1+L^2}{1-L^2}$? If these are the same then what is the advantage of your method? If there is a difference then what is the key difference here?
- Thm2: what is the relationship between $F$ and $G$? Is there a typo here?
- Sec3.4: what is the advantage of using concatenation?
- What is the complexity of your algorithms (log-det estimator, forward and back-prop)? How is it compared to previous methods especially ResFlow?
- What is the sampling speed of your method in experiments? What about quality evaluations such as FID? These numbers will make the experiment section more complete.


----------------------------------------------

After rebuttal:

Thanks for the author response. All my questions are answered. I think this is a very solid paper. I'll raise my score to 7.

**Limitations:**

Yes, they discussed limitations in the appendix.

**Strengths And Weaknesses:**

The paper is very clearly written. The application of monotone operators on invertible flows is novel to my knowledge. This paper is interesting to researchers who work on theory and applications of flow-based models, in the area of generative modeling.

Strengths:
- The biggest strength is that they derived $\mathcal{M}_L$ with the monotone formulation, and show this class is rigorously larger than the ones in i-ResNet or implicit NFs.
- The derivations of log-det estimator and back-prop are mostly standard. These provide ways to train their model with similar techniques as previous work (e.g. ResFlow).
- They introduced the new activation function called Pila. I am not an expert in activation functions but it seems to work well according to Table 3.
- They show better NLL than other flow-based models on several standard image datasets.

Weakness:
- As mentioned in Appendix F, their model has similar weaknesses as previous models such as computation and generation quality.
- In experiments, improvement on density estimation is very marginal, and cannot compete with other generative methods outside normalizing flows. For example, some score-matching based models achieved NLL<2.5 on CIFAR10.
- The experiments lack some other metrics such as sampling speed and FID of generated samples.

---

> ### Author Response · Authors · 2022-08-02
> **Response to Reviewer crL7**
>
> We would like to thank the reviewer for the positive assessment of our work and the insightful comments. During the revision, we have tried to address all comments, and we believe the paper has been further strengthened as a result. In the following, we provide our point-by-point response to the review comments:
>
> > Q1) Thm 4 says $\mathcal{M}_L = \frac{1+L^2}{1-L^2} \mathcal{R}_K$, $K = \frac{2L}{1+L^2}$. Since $K < 1$ when $L < 1$, is choosing a function from $\mathcal{G}_L$ and applying your formulation equivalent to choosing a function from $\mathcal{G}_K$ and applying from the residual formulation, if we disregard the constant $\frac{1+L^2}{1-L^2}$? If these are the same then what is the advantage of your method? If there is a difference then what is the key difference here?
>
> While $\mathcal{M}_L$ and $\mathcal{R}_K$ are equivalent theoretically, it is not possible, in practice, to take $L$ (or $K$) close to one. In fact, the variance of the log determinant estimator (6) diverges to infinity if $L$ exceeds a threshold $L_T < 1$ in the Russian roulette estimator. One of the advantages of our method is expressivity. For instance, with the same value of $L$, the residual formulation or the inverse residual formulation yields $\mathbb{R}^+ \mathcal{R}_L$, whereas our formulation yields $\mathbb{R}^+ \mathcal{R}_\frac{2L}{1+L^2}$ which is clearly more expressive than $\mathbb{R}^+ \mathcal{R}_L$. For more details, please refer to the explicit discussion in our paper, in Line 212-222.
>
> > Q2) Thm2: what is the relationship between $F$ and $G$? Is there a typo here?
>
> Yes, it is a typo. Thank you for pointing this out. The equation should be written as $C_F = 2(\mathrm{Id}+F)^{-1} - \mathrm{Id}$. We will fix the typo in the final version of the paper.
>
> > Q3) Sec3.4: what is the advantage of using concatenation?
>
> It helps to ameliorate the vanishing gradient problem, as suggested by i-DenseNets paper [2]. The vanishing gradient can be especially problematic for ResNet-based normalizing flows since they are subject to Lipschitz constraints. As in [2], which provided the experimental and theoretical analysis comparing LipSwish and concatenated LipSwish (CLipSwish), we have introduced Pila and concatenated Pila (CPila). Furthermore, we have provided a clear presentation showing that the CPila less attenuates the Lipschitz constant compared to CLipSwish [2], as visualized in Figure 2 (b) of our main paper.
>
> > Q4) What is the complexity of your algorithms (log-det estimator, forward and back-prop)? How is it compared to previous methods, especially ResFlow?
>
> Our log-determinant estimator has exactly the same complexity, since it is identical to that of Residual Flows (or i-DenseNets) except that the estimator is evaluated at $w = \left(\frac{Id + G}{2}\right)^{-1}(x)$ instead of at $x$. However, our forward and backward passes are both lengthened by fixed-point iterations. Overall, our model has about 60% longer training time compared to i-DenseNets, and 40% longer than Residual Flows. However, note that the sampling speed of our model is only 4% slower than Residual Flows. We will include the running time analysis in the final version of the paper.
>
> > Q5) What is the sampling speed of your method in experiments? What about quality evaluations such as FID? These numbers will make the experiment section more complete.
>
> Regarding the sampling speed, we have attached the time to sample a single batch of 100 images (trained on CIFAR-10) using a single RTX 3090 GPU. The numbers are averaged over ten batches. The full model takes approximately the same time as i-DenseNets, clearly demonstrating that the computational overhead induced by our method is negligible.
>
> | Model | Time (in seconds) |
> |------------------------------------|------------------------------|
> | i-DenseNets | 2.66 s |
> | i-DenseNets run with our implementation | 4.46 s |
> | Monotone Flows (ablate monotone formulation) | 4.59 s |
> | Monotone Flows (ablate CPila) | 2.58 s |
> | Monotone Flows (full model) | 2.70 s |
>
> Note: For fair comparison, both the benchmark option and the non-deterministic computation were disabled while obtaining the times.
>
> Regarding FID, we have attached the FID values for ours and related models. While our model is not free of the general tendency that normalizing flows do not excel at generating low-FID images, it does attain better FID values compared to the predecessors, especially Residual Flows and i-DenseNets.
>
> | Model | CIFAR-10 Train FID (lower is better) |
> |----------------|--------------------------------------|
> | PixelCNN | 65.93 |
> | PixelIQN | 49.46 |
> | i-ResNet | 65.01 |
> | Glow | 46.90 |
> | Residual Flows | 46.37 |
> | i-DenseNets | 41.29 |
> | Monotone Flows | 40.66 |
>
> [1] Cornish et al. Relaxing bijectivity constraints with continuously indexed normalising flows. In ICML, 2020.
>
> [2] Perugachi-Diaz et al. Invertible densenets with concatenated lipswish. In NeurIPS, 2021.
>
> [3] https://openreview.net/forum?id=btfPZRDdP2F

---

### Official Review · Reviewer_cueX · 2022-07-04

**Rating:** 7
**Confidence:** 4
**Soundness:** 3 good
**Presentation:** 3 good
**Contribution:** 3 good

**Summary:**

This paper proposes Monotone Flows which uses monotone operator theory to parameterize Residual Flows. The authors also introduce a novel activation function in Pila and CPila that improves upon the LipSwish activation function. As a theoretical contribution, the authors demonstrate the Monotone Flows using the Cayley operator are able to express a larger family of functions when compared to vanilla Residual and Inverse Residual Flows. Empirically the authors show the benefits of Monotone Flows over standard baselines in qualitative samples and density estimation.

**Questions:**

1. In the original Residual Flows paper there was a precise characterization of the memory cost (Fig 3). How does Krasnoselskii-Mann iteration affect this?
2. An effect of the activation through training like Fig 8 in the ResFlow paper could be illuminating. Can you please consider including it?

**Strengths And Weaknesses:**

Strengths
- The approach to parametrize Residual Flows using Monotone Operators and Cayley Operators is both an interesting and novel direction.
- The presented theory is both clear and integrates nicely with the overall goal of building a novel class of Normalizing Flow.
- The ablation experiments help justify the choice of the activation function.
- The experiments do enough to demonstrate that Monotone Flows are extremely expressive and powerful.
- Theorem 4, is especially crisp at outlining the main advantages of the Cayley Formulation from a theoretical point of view.

Weakness:
- One of the claims in the abstract is that Monotone Flows outperform SOTA flows. Looking at the presented numbers and comparing it with relevant literature this is not true. Here is a non-exhaustive list of papers that outperform Monotone Flows (albeit using other advances that could also potentially be complementarily used here): Flow++[1], Augmented Normalizing Flow [2], Densely Connected Normalizing Flows [3].
- The authors should also include FFJORD [4] as a relevant baseline.
- The use of Russian Roullete, Neumann series,  and Hutichinsons trace estimator for training and inference borrows much of what is already known in the flow literature. The authors should try to do a better job of disambiguating the main technical differences in Monotone Flows from the rest of the Residual Flow models.
- The reported tables should have standard deviations as the numbers are very close.


Minor:
- line 131, T hasn't been introduced yet.
- Definition 4, A is the set of what function? Be more precise.
- Why do Residual Flows have higher precision---i.e. one more digit after the decimal---in Table 2 for Cifar10?

[1] Ho, Jonathan, et al. "Flow++: Improving flow-based generative models with variational dequantization and architecture design." International Conference on Machine Learning. PMLR, 2019.

[2] Huang, Chin-Wei, Laurent Dinh, and Aaron Courville. "Augmented normalizing flows: Bridging the gap between generative flows and latent variable models." arXiv preprint arXiv:2002.07101 (2020).

[3] Grcić, Matej, Ivan Grubišić, and Siniša Šegvić. "Densely connected normalizing flows." Advances in Neural Information Processing Systems 34 (2021): 23968-23982.

[4] Grathwohl, Will, et al. "Ffjord: Free-form continuous dynamics for scalable reversible generative models." arXiv preprint arXiv:1810.01367 (2018).

---

> ### Author Response · Authors · 2022-08-02
> **Response to Reviewer cueX**
>
> We would like to thank the reviewer for the constructive feedback on our work and the insightful comments. We will address all the concerns raised by the reviewer below.
>
> > Q1) One of the claims in the abstract is that Monotone Flows outperform SOTA flows. Looking at the presented numbers and comparing it with relevant literature this is not true. Here is a non-exhaustive list of papers that outperform Monotone Flows (albeit using other advances that could also potentially be complementarily used here): Flow++[1], Augmented Normalizing Flow [2], Densely Connected Normalizing Flows [3].
>
> Yes, we acknowledge the comparison should have been made from a broader perspective, including Augmented Normalizing Flow which learns invertible transformations in an augmented data space. We appreciate your inspiring comment on the additional related works. We will tone down the SOTA claim and add the discussion about the suggested papers in the experiment section of the final version.
>
> > Q2) The authors should also include FFJORD [4] as a relevant baseline.
>
> We will include the work as a relevant baseline. Thanks for pointing this out.
>
> > Q3) The use of Russian Roulette, Neumann series, and Hutchinsons trace estimator for training and inference borrows much of what is already known in the flow literature. The authors should try to do a better job of disambiguating the main technical differences in Monotone Flows from the rest of the Residual Flow models.
>
> Thank you so much for your sincere comment on the presentation in the manuscript. In fact, we have already stated that the Russian Roulette estimator, Neumann series, and Hutchinsons trace estimator are due to prior works including [2], in Line 156 and Line 158. However, for clear presentation, we will add a sentence to our paper, stating that our key contributions are the monotone formulation and the activation function CPila, while the Russian Roulette estimator, Neumann series, and Hutchinsons trace estimator are adapted from prior work such as [2].
>
> > Q4) The reported tables should have standard deviations as the numbers are very close.
>
> We agree that running the experiment multiple times and reporting performance with standard deviations are ideal. However due to the limited computing resources and notoriously long training time of normalizing flows, previous works [1,2,3,4] reported their performance with a single run and it is de facto standard as Reviewer BzEz
> mentioned. In addition, we observe that training curves of i-DenseNets and Monotone Flows remain well-separated throughout the whole training session, which can be found in the updated supplementary material (.zip file). We will include the learning curves in the appendix of the final version, although Reviewer BzEz said a single run is a minor problem in the field of normalizing flows.
>
> > Q5) Line 131, $T$ hasn't been introduced yet.
>
> Yes, it is a typo. Thank you for pointing this out. Line 131 should be written as “Theorem 1 then implies $F$ is invertible.” (not $T$)
>
> > Q6) Definition 4, $A$ is the set of what function? Be more precise.
>
> Yes, $A$ could mean one of the function classes, $\mathcal{G}_L$, $\mathcal{R}_L$, $\mathcal{I}_L$, or $\mathcal{M}_L$. We will update the paper accordingly. Thanks for pointing this out.
>
> > Q7) Why do Residual Flows have higher precision---i.e. one more digit after the decimal---in Table 2 for Cifar10?
>
> We quoted the numbers as reported in the respective papers. The Residual Flows paper reports the numbers up to three decimal places, whereas all the other papers report the numbers up to two decimal places. In the final version of our paper, we will display all numbers up to two decimal places.
>
> > Q8) In the original Residual Flows paper there was a precise characterization of the memory cost (Fig 3). How does Krasnoselskii-Mann iteration affect this?
>
> We have conducted an ablation study on the code-level. The GPU memory consumption is decreased by about 1.3% when the fixed-point routines in the forward and backward passes are disabled while all other components are left intact.
>
> > Q9) An effect of the activation through training like Fig 8 in the ResFlow paper could be illuminating. Can you please consider including it?
>
> Thanks for the suggestion. We have included in the updated supplementary materials (.zip file) the training curves from the ablation study which compares the models with and without the activation function CPila. We will update the paper accordingly.
>
> [1] Behrmann et al. Invertible residual networks. In ICML, 2019.
>
> [2] Chen et al. Residual flows for invertible generative modeling. In NeurIPS, 2019.
>
> [3] Durk P. Kingma and Prafulla Dhariwal. Glow: Generative flow with invertible 1x1 convolutions. In NeurIPS, 2018.
>
> [4] Perugachi-Diaz et al. Invertible densenets with concatenated lipswish. In NeurIPS, 2021.
>
> [5] Lu et al. Implicit normalizing flows. In ICLR, 2021.

---

> > ### Comment · Reviewer_cueX · 2022-08-07
> > **Re:Rebuttal**
> >
> > Thank you for answering in detail to all my questions. While monotone operators are a fresh taking on normalizing flows I appreciate that the authors pledge to tone down the SOTA claim. I believe this is an interesting addition to the existing flow literature. Correspondingly, I'm increasing my original score from a 6->7.

---

### Official Review · Reviewer_ZJjf · 2022-07-08

**Rating:** 7
**Confidence:** 4
**Soundness:** 3 good
**Presentation:** 3 good
**Contribution:** 3 good

**Summary:**

The paper
* constructs normalizing flows based on monotone operators. Specifically, by Definition2, the paper defines $F(x)=(\frac{Id+G}{2})^{-1}(x) - x$ for a $L$-Lipschitz ($L<1$) function $G$ (thus $G=C_F$ is the Cayley operator of $F$). Next from the  Cayley operator identity $C_F+C_{F^{-1}} = 0$, $F^{-1}$ can be defined by
$$F^{-1}(x) = C_{-G}(x) = (\frac{Id-G}{2})^{-1}(x) - x.$$
For $G$, the authors use the architecture in i-DenseNets and propose a new activation function, CPila. For training and inference, log determinant can be computed via
$$\log \det J_F = \text{tr}[\log (I-J_G) - \log (I+J_G)]$$
and estimated with some existing unbiased estimators. Training and inference schemes are also developed well using existing techniques in this work.

* analyzes the expressive power of Monotone Flows theoretically, and compares with residual flows and implicit flows. Experimentally, the paper evaluates Monotone Flows on image density estimation tasks and shows the superiority to some normalizing flow models.





**Questions:**

* As mentioned in the weakness.B, some works are omitted in the experiments.

**Limitations:**

Limitations seen above. Societal impact: none

**Strengths And Weaknesses:**

**Strengths**:

A. To the best of my knowledge, constructing normalizing flows with monotone operator is novel.

B. Theoretic analysis of expressive power is valid. Experimental results also shows its superiority to some residual flows. Training and inference schemes based on existing techniques are also valid. Moreover, based on the observation that LipSwish suffers from the vanishing gradient problem, the paper proposes a new activation function CPila, which is also a novel contribution. Ablation studies also support the effects of CPila.

C. I think Monotone flows is a valid and novel contribution to the community of normalizing flow models.

**Weaknesses**:

A. Some notations in the paper are quite confusing. In Theorem2, the authors say $C_F = (\frac{Id+F}{2})^{-1} - Id$ is its Cayley operator, which is quite confusing and different from Line81. I think the definition in Line81 is the correct one and then the Definition2 is the 'inverse' Cayley operator so $G=C_F$.

B. The paper claims that they achieves the SOTA in density estimation task, but to my knowledge, there are some works not included in the experiment section such as Flow++ [1], VFlow [2], Scoreflow [3]

**Minor**:

**Reference**

[1] Ho, J., Chen, X., Srinivas, A., Duan, Y., and Abbeel, P. Flow++: Improving flow-based generative models with variational dequantization and architecture design. In International Conference on Machine Learning, pp. 2722– 2730, 2019.

[2] J. Chen, C. Lu, B. Chenli, J. Zhu, and T. Tian. Vflow: More expressive generative flows with variational data augmentation. In International Conference on Machine Learning, pages 1660–1669. PMLR, 2020.

[3] Song, Y., Durkan, C., Murray, I., and Ermon, S. Maximum likelihood training of score-based diffusion models. arXiv e-prints, pp. arXiv–2101, 2021.

---

> ### Author Response · Authors · 2022-08-02
> **Response to Reviewer ZJjf**
>
> First of all we would like to express our sincere gratitude for taking your time to review our paper and having a positive view on our proposal. We will try to address your questions below:
>
> > Q1) Some notations in the paper are quite confusing. In Theorem2, the authors say $C_F = \left(\frac{\mathrm{Id}+F}{2}\right)^{-1} - \mathrm{Id}$ is its Cayley operator, which is quite confusing and different from Line81. I think the definition in Line81 is the correct one and then the Definition 2 is the 'inverse' Cayley operator so $G = C_F$.
>
> Yes, in Theorem2, Line 118 has a typo. Thank you for pointing this out. The equation should be written as $C_F = 2(\mathrm{Id}+F)^{-1} - \mathrm{Id}$. We will fix this typo in the final version of our paper.
>
> > Q2) The paper claims that they achieve the SOTA in density estimation task, but to my knowledge, there are some works not included in the experiment section such as Flow++ [1], VFlow [2], Scoreflow [3].
>
> Yes, we acknowledge the comparison should have been made from a broader perspective, including Scoreflow, which is a hybrid of the diffusion model and normalizing flows. We appreciate your inspiring comment on the additional related works. We will tone down the SOTA claim and discuss the suggested papers in the experiment section of the final version.

---

### Official Review · Reviewer_BzEz · 2022-07-11

**Rating:** 7
**Confidence:** 4
**Soundness:** 3 good
**Presentation:** 4 excellent
**Contribution:** 3 good

**Summary:**

The paper proposes a new type of ResNet-based Normalizing Flows. In contrast to prior studies, which required the Lipschitz constant $L$ of each layer to be less than 1, the authors use monotone operators, which they show to be strictly more expressive. A new activation function called Concatenated Pila (CPila) is also proposed. The suggested model is evaluated on multiple toy datasets as well as standard image datasets, outperforming the baseline i-DenseNet model.

**Questions:**

* If possible with time, it would be nice if the authors could run their model with variational dequantization on CIFAR10 for the rebuttal.
* Regarding monotone and maximally monotone operators: On line 79, what is meant by $u \in F(x), v \in F(y)$? From my understanding, $F(x)$ is an $n$-dimensional vector and not a set, so how can it have elements? The same applies on line 108: Given that $F$ is a function, what does it mean for $F$ not to be a proper *subset* of any monotone operator?
* Line 118: What is $G$ in the definition of $C_F$? Should there be an $F$ in the numerator?


**Limitations:**

Yes, the limitations are adequately addressed.

**Strengths And Weaknesses:**

### Strengths

**Originality**

The formulation of monotone operators is something I haven’t encountered previously in the field. The concept seems intriguing and well-conceived.

**Quality**

I did not check the proofs in the appendix, but the mathematical theory in the main part is sound and comprehensible. The experiments and ablations follow the established standard of the field.

**Clarity**

The paper is very clearly written and easy to read. The authors do a good job at separating the main mathematical theory from the details and the proofs which are presented in the appendix.

**Significance**

Given the originality of the monotone formulation, the paper may inspire future work in the field of normalizing flows. The reported improvements are small, but consistent over multiple datasets.

### Weaknesses

* The paper says that *The resulting model, Monotone Flows, exhibits an excellent density estimation performance and outperforms existing state-of-the-art normalizing flow models on multiple density estimation benchmarks (MNIST, CIFAR-10, ImageNet32, ImageNet64).* This is true when uniform dequantization is used during training. However, the Flow++ paper ([34]) reports an even lower bits-per-dimension score when using variational dequantization. Therefore, the proposed model should also be trained with variational dequantization and compared to Flow++ before making the state-of-the-art claim.
* Some parts of the definition of monotone and maximally monotone operators are a bit confusing. (Further details in the Questions section below.)
* Each of the experiment was only run a single time, which makes it harder to assess the significance of the reported numbers. However, this is a standard practice in the field of normalizing flows because each training run can take a lot of time. So this is only a minor problem, given the state of the field.

---

> ### Author Response · Authors · 2022-08-02
> **Response to Reviewer BzEz**
>
> First of all we would like to express our sincere gratitude for taking your time to review our paper and having a positive view on our proposal. We will try to address all of your concerns below:
>
> > Q1) The paper says that The resulting model, Monotone Flows, exhibits an excellent density estimation performance and outperforms existing state-of-the-art normalizing flow models on multiple density estimation benchmarks (MNIST, CIFAR-10, ImageNet32, ImageNet64). This is true when uniform dequantization is used during training. However, the Flow++ paper ([34]) reports an even lower bits-per-dimension score when using variational dequantization. Therefore, the proposed model should also be trained with variational dequantization and compared to Flow++ before making the state-of-the-art claim.
>
> Yes, we acknowledge that the comparison should have been made from a broader perspective. We appreciate your inspiring comment on variational dequantization. We will tone down the SOTA claim in the final version of our paper.
>
> > Q2) Each of the experiments was only run a single time, which makes it harder to assess the significance of the reported numbers. However, this is a standard practice in the field of normalizing flows because each training run can take a lot of time. So this is only a minor problem, given the state of the field.
>
> We agree that it is ideal to run experiments multiple times in order to more rigorously assess the significance of improvement. However, as you have mentioned, normalizing flows usually take a significant amount of time to train, and it is a standard practice to report single-run results [1,2,3,4]. Nevertheless, we would like to note that the training curves of ablated models remain well-separated from the training curve of the full model throughout the whole training session, which can be checked in the updated supplementary material (.zip file).
>
> > Q3) If possible with time, it would be nice if the authors could run their model with variational dequantization on CIFAR10 for the rebuttal.
>
> We would like to run our model with varitional dequantization but unfortunately, training normalizing flows takes a significant amount of time. We are sorry that it would not be possible to run additional experiments due to the limited time period.
>
> > Q4) Some parts of the definition of monotone and maximally monotone operators are a bit confusing. Regarding monotone and maximally monotone operators: On line 79, what is meant by $u \in F(x)$, $v \in F(y)$? From my understanding, $F(x)$ is an $n$-dimensional vector and not a set, so how can it have elements? The same applies on line 108: Given that $F$ is a function, what does it mean for not to be a proper subset of any monotone operator?
>
> Thank you for the careful reading. In general, the function $F$ is considered as a single-valued function defined at every point. However, in monotone operator theory, we tried to be more general by allowing $F$ to be an “operator”, which is a subset of $\mathbb{R}^n\times\mathbb{R}^n$, and may have some points where F is undefined or has multiple values. Following mathematical convention, we overload the notation $F(x)$ to mean the set $F(x) = \lbrace y | (x,y) \in F \rbrace$. Please refer to [5, section 2.1] for more details.
>
> > Q5) Line 118: What is $G$ in the definition of $C_F$? Should there be an $F$ in the numerator?
>
> It is a typo. The equation should be written as $C_F = 2(\mathrm{Id}+F)^{-1} - \mathrm{Id}$. We will correct the typo in the final version.
>
> [1] Jens Behrmann, Will Grathwohl, Ricky T. Q. Chen, David Duvenaud, and Jörn-Henrik Jacobsen. Invertible residual networks. In ICML, 2019.
>
> [2] Ricky T. Q. Chen, Jens Behrmann, David K. Duvenaud, and Jörn-Henrik Jacobsen. Residual flows for invertible generative modeling. In NeurIPS, 2019.
>
> [3] Durk P. Kingma and Prafulla Dhariwal. Glow: Generative flow with invertible 1x1 convolutions. In NeurIPS, 2018.
>
> [4] Yura Perugachi-Diaz, Jakub Tomczak, and Sandjai Bhulai. Invertible densenets with concatenated lipswish. In NeurIPS, 2021.
>
> [5] Ryu, E., & Yin, W. (2022). Large-Scale Convex Optimization: Algorithm Analysis via Monotone Operators. Cambridge: Cambridge University Press.

---

> > ### Comment · Reviewer_BzEz · 2022-08-08
> > **Reply to Rebuttal**
> >
> > Thank you for your reply. I'm happy with all the answers.
> >
> > I would suggest to incorporate the reply to Q4 into the paper (i.e. that the notation is overloaded and that $F(x) = \\{y \; | \; (x, y) \in F\\}$). This would help to avoid confusing authors without a background in monotone operator theory.
> >
> > After reading the other reviews and replies, I keep my rating at 7 and recommend the acceptance of the paper.

---

### Author Response · Authors · 2022-08-02
**Overall response**

We would like to thank you for handling the review of our manuscript entitled *“Invertible Monotone Operators for Normalizing Flows”* and for unanimous support on the novelty of our work. During the revision, we have addressed all suggestions/comments that have been made in the review. Also, we have updated our supplementary material (.zip file) for an additional figure.

---

### Meta-Review · Area_Chair_Y2hE · 2022-08-23

**Recommendation:** Accept
**Confidence:** Certain

**Metareview:**

The paper proposes a new type of ResNet-based Normalizing Flows that, unlike previous versions of these flows, do not require the Lipschitz constant of each layer to be less than 1. The authors use monotone operators, which they show to be strictly more expressive and propose a new activation function called Concatenated Pila (CPila). The method is evaluated on toy datasets as well as standard image datasets, outperforming the baseline i-DenseNet model.

Strengths:

1 - Well written and clear paper.
2 - Originality of the monotone formulation.
3 - Improvements are small, but consistent across multiple settings.
4 - Theoretical analysis of expressive power.
5 - Ablation experiments to justify activation function.

Weaknesses:

- No significant weaknesses are mentioned by the reviewers.

Decision:

All the reviewers agree on acceptance, indicating that this is a strong paper. I encourage the authors to use the feedback provided by the reviewers to improve the paper for its camera-ready version.

**Award:**

No

---

### Decision · Program_Chairs · 2022-09-14

Accept